# A role for Tau protein in maintaining ribosomal DNA stability and cytidine deaminase-deficient cell survival

Elias Bou Samra[1,2,3], Géraldine Buhagiar-Labarchède [1,2,3], Christelle Machon[4,5], Jérôme Guitton[4,6], Rosine Onclercq-Delic [1,2,3], Michael R. Green [7], Olivier Alibert [8], Claude Gazin[8], Xavier Veaute [9] & Mounira Amor-Guéret [1,2,3]

Cells from Bloom's syndrome patients display genome instability due to a defective BLM and the downregulation of cytidine deaminase. Here, we use a genome-wide RNAi-synthetic lethal screen and transcriptomic profiling to identify genes enabling BLM-deficient and/or cytidine deaminase-deficient cells to tolerate constitutive DNA damage and replication stress. We found a synthetic lethal interaction between cytidine deaminase and microtubule-associated protein Tau deficiencies. Tau is overexpressed in cytidine deaminase-deficient cells, and its depletion worsens genome instability, compromising cell survival. Tau is recruited, along with upstream-binding factor, to ribosomal DNA loci. Tau downregulation decreases upstream binding factor recruitment, ribosomal RNA synthesis, ribonucleotide levels, and affects ribosomal DNA stability, leading to the formation of a new subclass of human ribosomal ultrafine anaphase bridges. We describe here Tau functions in maintaining survival of cytidine deaminase-deficient cells, and ribosomal DNA transcription and stability. Moreover, our findings for cancer tissues presenting concomitant cytidine deaminase underexpression and Tau upregulation open up new possibilities for anti-cancer treatment.

[1] Institut Curie, PSL Research University, UMR 3348, Orsay 91405, France. [2] CNRS UMR 3348, Centre Universitaire, Orsay 91405, France. [3] Université Paris Sud, Université Paris-Saclay, Centre Universitaire, UMR 3348, Orsay 91405, France. [4] Laboratoire de Biochimie et Toxicologie, Hospices Civils de Lyon, Centre Hospitalier Lyon-Sud, Pierre-Bénite 69495, France. [5] ISPB Faculté de Pharmacie, Laboratoire de Chimie Analytique, Université de Lyon, Université Lyon 1, Lyon 69008, France. [6] ISPB, Faculté de PharmacieLaboratoire de Toxicologie, Université de Lyon, Université Lyon 1, Lyon 69008, France. [7] Howard Hughes Medical Institute, University of Massachusetts Medical School, Worcester, Massachusetts 01605, USA. [8] CEA-DRF-iRCM-LEFG-Genopole, Evry 91057, France. [9] CEA-DRF-iRCM-CIGEx, Fontenay-aux-Roses 92265, France. Correspondence and requests for materials should be addressed to M.A.-G. (email: mounira.amor@curie.fr)

Every life form delivers its genetic material to the next generation. However, a myriad of alterations can undermine the integrity of this process, thereby favoring genomic instability, which can drive diseases, premature aging and tumorigenesis[1].

Cells from Bloom's syndrome (BS) patients display high levels of genomic instability. BS belongs to a group of rare human genetic diseases with a particularly high rate of spontaneous chromosome abnormalities[2, 3]. BS results from mutations of both copies of the *BLM* gene, which encodes a 3′–5′ DNA helicase[4] and is characterized by a high incidence of sister chromatid exchanges[2, 4, 5] and strong predisposition to cancers[6]. BS cells suffer from replication stress and chromosome segregation defects, including an abnormally high frequency of ultrafine anaphase bridges (UFBs). We have shown that BLM deficiency leads to the downregulation of cytidine deaminase (CDA), an enzyme of the pyrimidine salvage pathway[7]. CDA catalyzes the hydrolytic deamination of cytidine (C) and deoxycytidine (dC) to uridine (U) and deoxyuridine (dU), respectively[8]. The imbalance in the nucleotide pool resulting from the CDA defect, either in BLM-deficient BS cells or BLM-proficient HeLa cells, reproduced several aspects of the genetic instability associated with BS condition[7, 9]. These data suggest that BS cells lacking both BLM and CDA, and CDA-deficient HeLa cells have developed mechanisms for tolerating endogenous DNA damage and replication stress.

In this study, we aimed to identify interactors enabling BLM-deficient and/or CDA-deficient cells to survive despite constitutive genetic instability, thereby contributing to carcinogenesis. We performed a genome-wide shRNA screen with a BS cell line, and its counterpart in which BLM function was corrected. The BS cells were likely to display higher levels of cell lethality due to the depletion of the microtubule-associated protein Tau. This lethality was observed in various CDA-deficient cells, but not in BLM-deficient cells expressing CDA, revealing a synthetic lethal interaction between Tau and CDA deficiencies.

Multiple functions have been attributed to Tau, based on its broad distribution within cells. In particular, nuclear Tau was shown to preserve DNA integrity in neurons, under both physiological and DNA-damaging conditions[10, 11]. Here, we observe the corecruitment of Tau and upstream binding factor (UBTF) to the nucleolar organizing regions (NORs), and find that Tau silencing reduces the recruitment of UBTF to ribosomal DNA (rDNA) repeats, thereby impairing rDNA transcription. Tau depletion also associates with lower intracellular ribonucleotide concentrations, consistent with the observed decrease in rDNA transcription. Moreover, the staining pattern for mitotic Tau foci reveals the presence of a new class of human UFBs extending from rDNA repeats. These rDNA-associated UFBs are particularly abundant in situations of nucleotide pool distortion and replication challenge. Finally, Tau depletion is sufficient to cause genomic instability, and its coupling with CDA deficiency aggravates this instability. These results reveal a function for Tau in rDNA metabolism, and indicate that Tau is critical for the survival of CDA-deficient cells, through its contribution to the safeguarding of genome integrity.

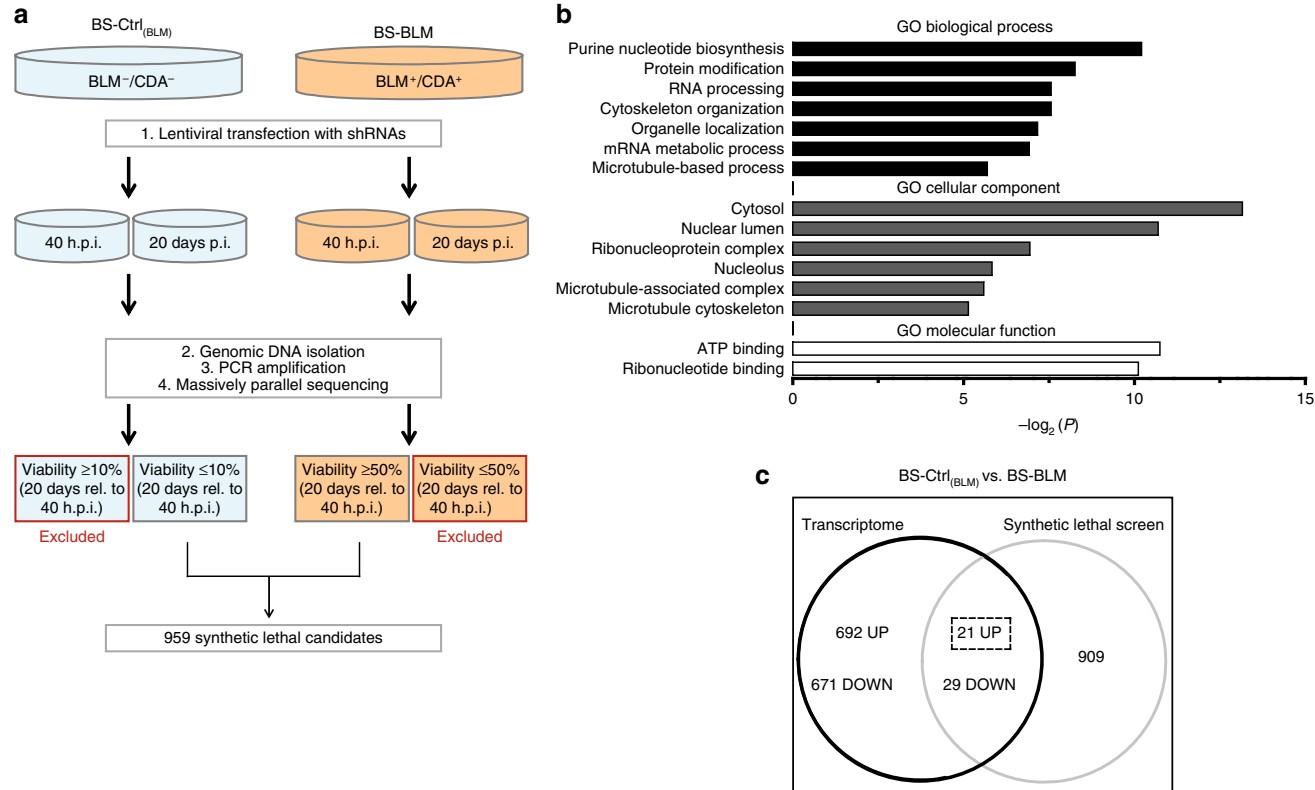

**Fig. 1** RNAi-based synthetic interaction screen. **a** Schematic representation of the genome-wide shRNA-screening procedure. Briefly, the screening involved four steps: (1) transduction of BS-Ctrl(BLM) and BS-BLM cells with a library of pooled lentiviral human shRNAs; (2) isolation of genomic DNA from cell populations 40 h or 20 days post-infection (*p.i.*); (3) PCR amplification of isolated genomic DNA; (4) quantification of shRNA populations by barcode sequencing. **b** Identification of pathway enrichment for the list of 959 synthetic lethal genes, with the DAVID database. The *x* axis corresponds to $-\log_2(P)$. *GO* Gene Ontology. **c** Venn diagram showing mRNAs identified as differentially expressed in BS-Ctrl(BLM) and BS-BLM cells, and the number of mRNAs concomitantly synthetic lethal

## Results

**RNAi-synthetic interaction screen in BS cells.** We searched for genes potentially required for the viability and proliferation of BS cells, by conducting a genome-wide RNAi screen with a human shRNA library comprising ~60,000 shRNAs directed against ~27,000 human genes[12]. We screened an isogenic pair of GM8505B-derived BS cell lines in parallel. The first line lacked the BLM protein and therefore displayed strong downregulation of CDA expression (BS-Ctrl$_{(BLM)}$, BLM$^-$/CDA$^-$), whereas the helicase defect of the second line was functionally corrected by stable transfection with the BLM cDNA, which also restored CDA expression (BS-BLM, BLM$^+$/CDA$^+$) (Supplementary Fig. 1).

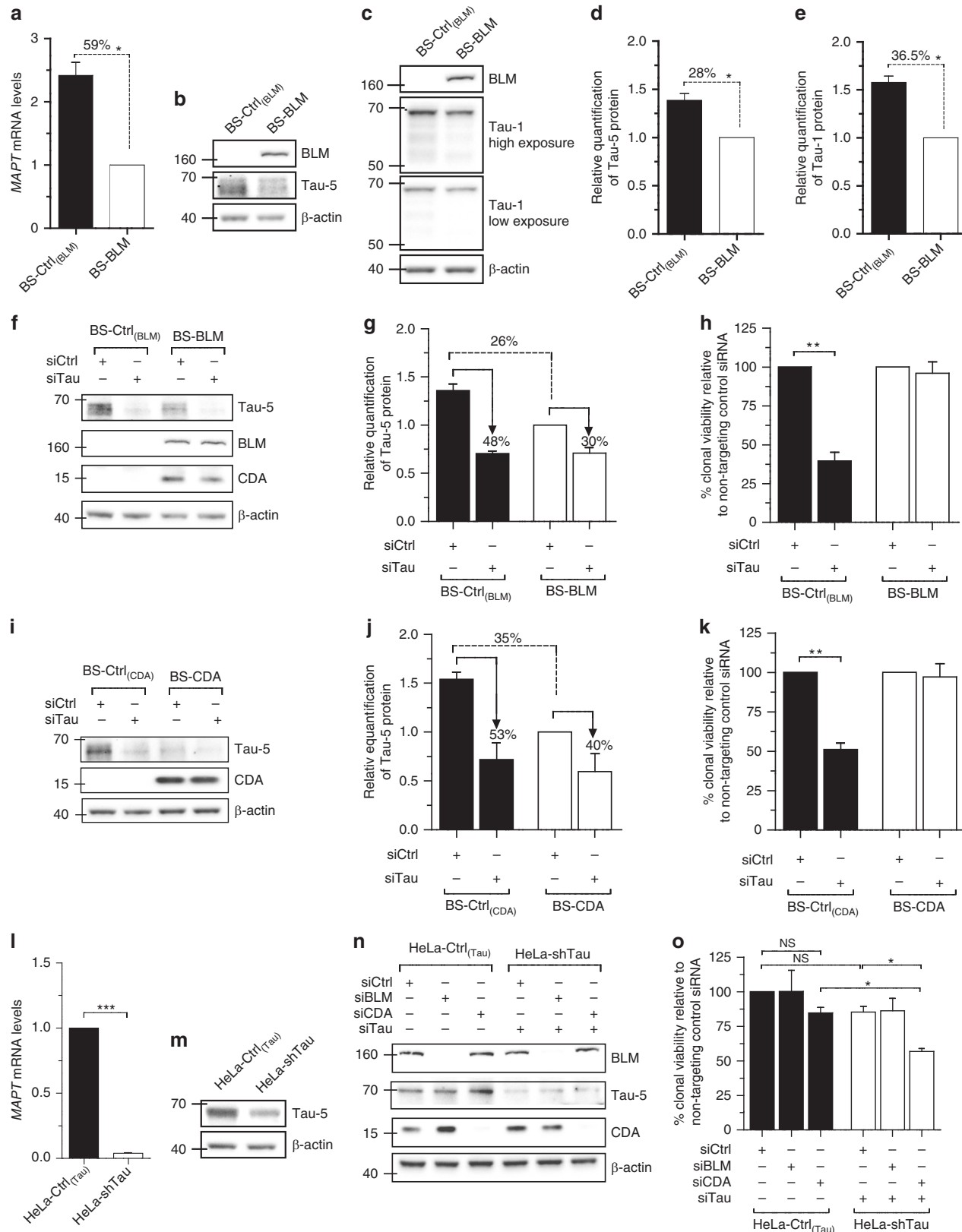

This screen might, therefore, allow the identification of genes required for the survival of BLM-deficient and/or CDA-deficient cells. An overview of the workflow is summarized in Fig. 1a. We identified 959 targets in our primary screen as candidate synthetic lethal genes in BS cells (Supplementary Data 1).

Gene ontology (GO) analyses of hits with DAVID[13] revealed enrichment in a group of 12 terms associated with nucleotide binding (Enrichment score = 2.86, Supplementary Table 1). The most significant GO biological process term was "purine nucleotide biosynthesis" ($P = 8.79 \times 10^{-4}$, $-\log_2(P) = 10.15$; Modified Fisher exact test; Fig. 1b, Supplementary Table 2). The importance of this cellular process was strongly reflected in the GO molecular function terms identified as enriched, which included ATP binding ($P = 5.84 \times 10^{-4}$, $-\log_2(P) = 10.74$; Modified Fisher exact test) and ribonucleotide binding ($P = 9.05 \times 10^{-4}$, $-\log_2(P) = 10.11$; Modified Fisher exact test; Fig. 1b, Supplementary Table 3). Other DAVID GO terms associated with microtubule cytoskeleton and organization were among the top clusters observed in the group of synthetic lethal genes ($P = 5.31 \times 10^{-3}$, $-\log_2(P) = 7.56$; Modified Fisher exact test; Fig. 1b, Supplementary Table 2). The identification of these genes presumably reflects the essential role of many cytoskeletal proteins in BS cell viability.

We limited the number of candidates in our primary screen, by cross-referencing these data with a list of genes upregulated in BS-Ctrl$_{(BLM)}$ relative to BS-BLM cells. We studied mRNA levels with the Affymetrix U133 Plus 2.0 Arrays previously used for these cell lines[7] (refer to the Methods section for link). Significant differences in mRNA levels were detected, with an absolute expression ratio difference of $\geq 1.5$ and a $P < 0.05$ (Moderated $t$-test). On the basis of these criteria, 692 genes were considered to be upregulated in BS cells relative to controls (Fig. 1c, Supplementary Data 2). Twenty-one of these genes (3%) were identified in the synthetic lethal screen (Supplementary Table 4). GO analysis was performed on this group of genes. Further enrichment in the biological process term "regulation of cytoskeleton organization" ($P = 8.44 \times 10^{-3}$, enrichment = 19.89; Modified Fisher exact test) and the cellular component term "cytoskeletal part" ($P = 3.33 \times 10^{-2}$, enrichment = 4.88; Modified Fisher exact test, Supplementary Table 5) was observed. Three genes corresponding to these enriched terms were highly represented: *CCDC88A*, *CAPG*, and *MAPT* (Supplementary Table 5). *CCDC88A* and *CAPG* encode actin-binding proteins that play a role in cytoskeleton remodeling and cell migration[14–16]; *MAPT* encodes Tau, a microtubule-binding protein that stimulates microtubule assembly by polymerizing with tubulin, thereby participating in cytoskeleton organization and integrity[17]. Strong expression of these three genes has been observed in various tumors, and has been reported to promote metastasis and increase the invasiveness of tumor cells[18–20].

In this study aiming to identify key players favoring the tolerance of constitutive genetic instability in BS cells, we searched for specific cytoskeletal genes likely to favor the maintenance of genome integrity. *MAPT* was the only one of these three genes already known to have a DNA-protecting function. Tau has been shown to protect neuronal nuclear DNA from heat stress-induced damage, and to promote chromosome stability[10, 11, 21]. We thus focused on Tau in subsequent experiments.

**Tau silencing reduces the survival of CDA-deficient cells.** In our transcriptomic data, Tau mRNA levels in BS cells were found to be twice those in controls (Supplementary Table 4). By examining microarray expression data for BLM-depleted or non-specific shRNA-treated isogenic control fibroblasts available under accession no. GSE54502[22], we also found that Tau was significantly upregulated in BLM-depleted cells relative to control cells (Fold-change = 1.45, $P = 7.63 \times 10^{-4}$; Moderated $t$-test; Supplementary Fig. 2a, b), supporting our findings. The over-expression of Tau in BS cells was confirmed by reverse transcription quantitative PCR (RT-qPCR) and western blotting (WB) (Fig. 2a–c). We used Tau-5 monoclonal antibody, which detects all Tau proteins, regardless of their phosphorylation state (Fig. 2b). As we planned to investigate the nucleolar distribution of Tau in subsequent experiments and the Tau-5 antibody is not suitable for Tau nucleolar immunostaining, we also tested the only Tau antibody for which nucleolar staining has been reported[23], the Tau-1 antibody (Fig. 2c). This antibody recognizes the unphosphorylated Tau epitope Ser195-202. BLM expression reduced Tau mRNA levels by 59%, (Fig. 2a) and the amounts of protein detected by Tau-5 and Tau-1 by 28 and 36.5%, respectively (Fig. 2d, e).

For validation of the apparent synthetic lethality interaction between Tau and BLM and/or CDA deficiencies, we transfected BS-Ctrl$_{(BLM)}$ and BS-BLM cells with a pool of four Tau-targeting siRNAs. Then, 72 h after transfection, we assessed changes in *MAPT* mRNA and Tau protein levels in BS-Ctrl$_{(BLM)}$ and BS-BLM cells by RT-qPCR and WB, respectively (Fig. 2f, g and Supplementary Fig. 2c–e). Both cell lines were monitored for colony formation. Tau downregulation in BLM-complemented cells did not affect the clonal growth (Fig. 2h). However, a 45–48% decrease in Tau protein levels resulted in a 60% decrease in the colony-forming ability of BLM-null cells (Fig. 2g, h and Supplementary Fig. 2e), confirming the observations made in the primary shRNA screen.

BLM deficiency decreases CDA levels in BS cells[7]. We investigated whether the synthetic lethal phenotype observed on Tau downregulation was due to CDA deficiency per se. We performed colony formation assays in BS-CDA (BLM⁻/CDA⁺) and CDA-deficient cells (BS-Ctrl$_{(CDA)}$, BLM⁻/CDA⁻) after Tau

**Fig. 2** Tau silencing reduces the survival of CDA-deficient cells. **a** *MAPT* mRNA levels determined by RT-qPCR in BS-Ctrl$_{(BLM)}$ and BS-BLM cells. **b**, **c** BLM and Tau levels determined by western blotting in BS-Ctrl$_{(BLM)}$ and BS-BLM cells using BLM and either Tau-5 (**b**) or Tau-1 (**c**) antibodies. **d**, **e** Tau protein quantification for Tau-5 (**d**) and Tau-1 (**e**) immunoreactivity. The β-actin signal was used as the control. The results are normalized against those for BS-BLM, which were set to 1. **f–k** Cells were first transfected with either non-targeting or Tau siRNA. After 24 h, cells were again transfected with the indicated siRNAs. Two days after the second round of transfection, the cells were collected for western blotting **f**, **i**, or were plated, in a serial dilution series, in 12-well plates. Ten to 12 days later, colonies were fixed and stained with crystal violet. Non-targeting siRNA was used as a control, with values set to 1 **h**, **k**. **g**, **j** Tau protein quantification after Tau-targeting siRNA depletion. **l** *MAPT* mRNA and **m** Tau protein levels determined in HeLa-Ctrl$_{(Tau)}$ and HeLa-shTau cells. **n**, **o** HeLa-Ctrl$_{(Tau)}$ and HeLa-shTau cells were first transfected with the indicated siRNAs. After 24 h, cells were again transfected with the same siRNAs. Two days after the second round of transfection, cells were collected for western blotting (**n**), or were plated, in a serial dilution series, in 12-well plates. Seven days later, colonies were fixed and stained with crystal violet. Non-targeting siRNA was used as a control, with the values obtained set to 1 (**o**). For qPCR, the mean value for 3 independent experiments is represented as a fold-change. *B2M*, *β-actin*, and *TBP* were used as housekeeping genes for qPCR normalization. For western blots, β-actin was used as a loading control. Each data bar is the mean of at least three independent experiments performed in triplicate. *Error bars* represent ± SEM. The significance of differences was assessed in two-tailed paired Student's $t$-tests. ***$P < 0.0005$, **$P < 0.005$, *$P < 0.05$, *NS* not significant

depletion (Fig. 2i–k and Supplementary Fig. 2f, g). The stable expression of CDA in BS cells reduced Tau protein levels by 35 and 39%, as shown with the Tau-5 and Tau-1 antibodies, respectively (Fig. 2i, j and Supplementary Fig. 2f, g), suggesting that high levels of Tau expression in BS cells probably results from CDA deficiency rather than from BLM deficiency per se.

CDA expression in BLM-deficient cells was sufficient to rescue the inhibition of clonal growth (Fig. 2k). Thus, it is CDA depletion, rather than a lack of BLM, that induces synthetic lethality in the absence of Tau. For confirmation of these results, we knocked down Tau levels in HeLa cells (HeLa-shTau) (Fig. 2l, m and Supplementary Fig. 2h). Tau depletion had no

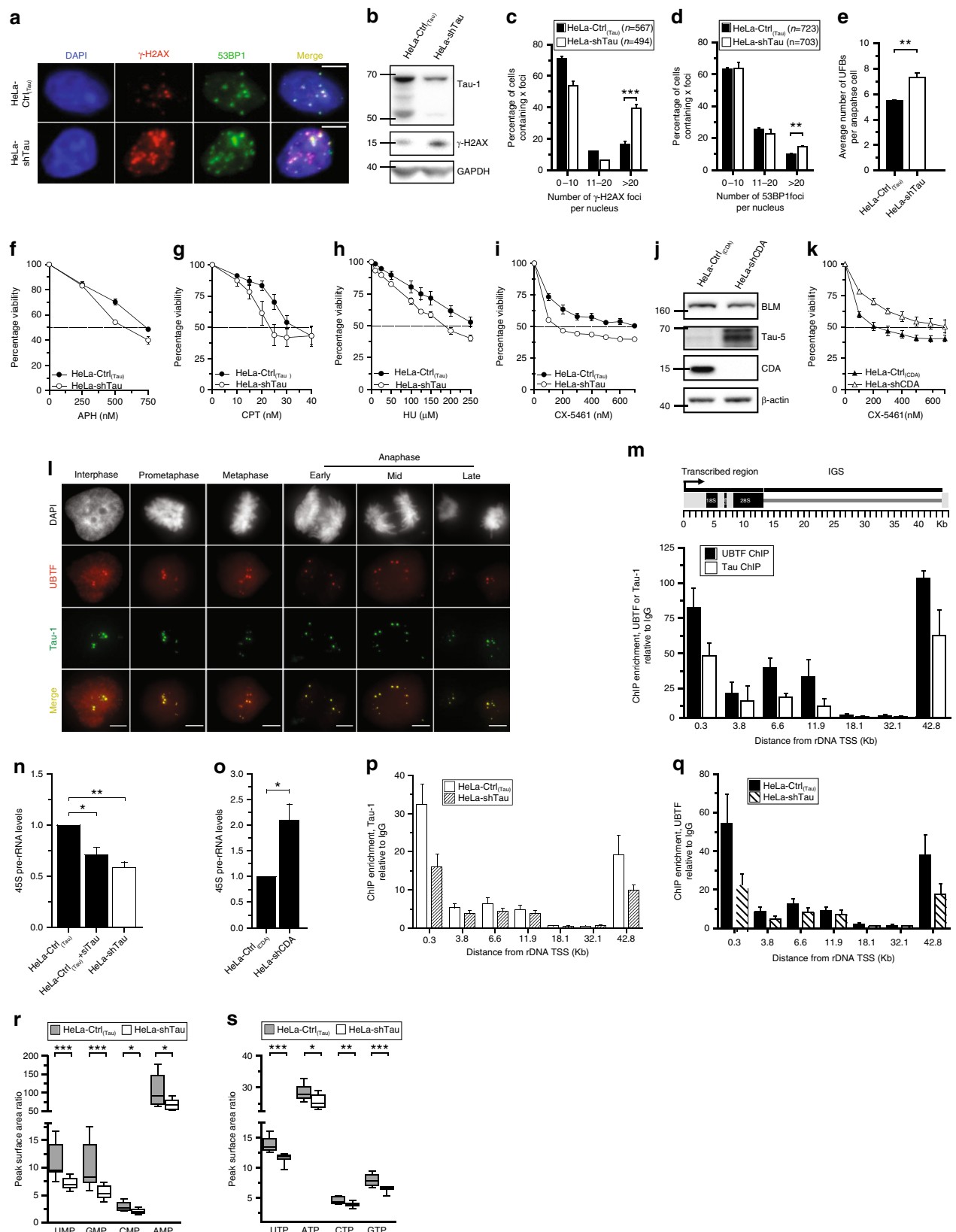

impact on the clonal survival of either of the isogenic cell lines (Supplementary Fig. 2i). In these cells, BLM or CDA levels were downregulated with specific siRNAs (Fig. 2n). We also transiently depleted Tau from HeLa-shTau cells with a pool of siRNAs directed against sequences other than that targeted by the shRNA construct, to obtain better extinction of Tau mRNA and protein levels (Supplementary Fig. 2j, k and Fig. 2m). An analysis of colony-forming ability showed that the silencing of CDA, but not of BLM, impaired the clonal growth of Tau-deficient cells by ~30% relative to controls (Fig. 2o). Our data demonstrate that Tau depletion is synthetic lethal with CDA deficiency.

**Tau interacts with rDNA and regulates rDNA transcription.** To decipher the molecular basis of the synthetic lethal interaction between CDA and Tau deficiencies, we investigated the potential role of Tau in maintaining genome stability in non-neuronal cells. We thus analyzed the impact of Tau knockdown on phosphorylated H2AX (γ-H2AX) levels, 53BP1 focus formation and UFB frequency (Fig. 3a–e). Tau silencing increased γ-H2AX levels, as revealed by WB and the high percentage of cells with more than 20 γ-H2AX foci (Fig. 3b, c). Tau knockdown also generated a significantly higher frequency of 53BP1 foci at nuclei, and resulted in a significant increase in UFB frequency, consistent with the existence of endogenous DNA damage and replication stress in Tau-deficient cells (Fig. 3d, e).

Based on previous reports indicating a role for nuclear Tau protein in neuronal DNA and RNA protection[10, 11], we assessed the sensitivity of non-neuronal parental and Tau-deficient HeLa cells to various cytotoxic agents in viability assays. Tau-deficient cells were as sensitive as parental cells to the catalytic inhibitor of topoisomerase II ICRF-159 or the inhibitor of RNA polymerase II α-amanitin (Supplementary Fig. 3a, b), 72 h after treatment. However, Tau silencing resulted in mild sensitivity to the B-family DNA polymerases inhibitor aphidicolin (APH), the topoisomerase I inhibitor camptothecin (CPT) and the ribonucleotide reductase inhibitor hydroxyurea (HU) (~1.5-fold; Fig. 3f–h), and a marked increase in sensitivity to the specific RNA polymerase I inhibitor CX-5461 (~3.5-fold; Fig. 3i), suggesting a role for Tau in ribosomal RNA (rRNA) transcription. As described above, CDA-deficient BS cells overexpressed Tau. Consistent with this finding, HeLa cells with physiological levels of BLM but a stable depletion of CDA also overexpressed Tau (Fig. 3j). We subjected these cells to drug sensitivity assays. CDA-deficient cells and control cells had similar sensitivities to ICRF-159, α-amanitin, and APH (Supplementary Fig. 3c–e). After treatment with CPT or HU, CDA-deficient cells became slightly less or more sensitive than control cells, respectively

(~1.5-fold; Supplementary Fig. 3f, g). By contrast, CDA depletion led to a 3.5-fold increase in resistance to CX-5461 (Fig. 3k), probably due to Tau overexpression. This greater resistance of CDA-deficient cells to CX-5461 treatment was inversely correlated with the greater susceptibility of Tau-depleted cells to the same treatment. Collectively, these findings provide further support for a role for Tau in rRNA transcription.

Within the nucleus, Tau has been shown to localize specifically to the fibrillary regions of interphase nucleoli and the NORs[23, 24]. We examined Tau nucleolar localization by the co-immunostaining of BS cells for the RNA polymerase I regulatory protein UBTF. UBTF is a key component of the pre-initiation complex (PIC) of rRNA transcription, mediating the recruitment of RNA polymerase I to rDNA promoter regions[25]. We found that, during interphase, Tau was co-localized with UBTF, in discrete foci within the nucleolus. When the cells entered prometaphase, Tau and UBTF formed paired foci linked to the NORs on acrocentric chromosomes. These foci persisted throughout prophase and metaphase, subsequently segregating symmetrically during early, mid and late anaphase (Fig. 3l). The Tau-1 antibody efficiently labels nucleolar mitotic Tau by detecting an unphosphorylated Tau epitope, but it is widely accepted that Tau phosphorylation increases during mitosis in neuronal and non-neuronal cells[26]. We therefore examined interphase and mitotic BS cells by immunofluorescence microscopy with two phosphorylation-dependent antibodies sensitive to phosphorylation of the Thr231 and Ser396 epitopes. As previously reported, very low levels of Tau phosphorylation were detected in interphase BS cells, whereas Tau became highly phosphorylated during mitosis (Supplementary Fig. 3h, i). In mitotic cells, immunoreactivity with the P-Thr231 antibody was observed at centrosomes during prometaphase. This immunoreactivity decreased during metaphase and anaphase, and was seen again in the midbody region during telophase (Supplementary Fig. 3h). Immunoreactivity with the P-Ser396 antibody was also localized to the midbody region in anaphase and telophase cells (Supplementary Fig. 3i). Thus, in BS cells, Tau displays low levels of phosphorylation in interphase cells, but becomes highly phosphorylated during mitosis, as reported in neuronal cells[26]. However, none of the tested phosphorylation-dependent antibodies localized to the nucleolus or to the NORs.

The co-localization of Tau with UBTF to the NORs led us to investigate whether Tau interacted with rDNA. We performed chromatin immunoprecipitation (ChIP)-qPCR with primers covering the different regions of the rDNA gene (Fig. 3m). One repeat contains ~13.3 kb of sequence encoding the 18 S, 5.8 S, and 28 S rRNAs and a non-coding intergenic spacer (IGS) containing an enhancer, a spacer promoter and the core promoter of the

**Fig. 3** Tau interacts with rDNA and regulates rDNA transcription. **a** Immunofluorescence microscopy showing γ-H2AX (*red*) and 53BP1 (*green*) labeling. Merged images in the right panel. *Scale bar*: 5 μm. **b** γ-H2AX and Tau-1 protein levels assessed by immunoblotting. GAPDH was used as a loading control. **c**, **d** Cells were immunostained with antibodies against γ-H2AX and 53BP1. At least 400 cells were acquired, and foci counted. The number of cells (*n*) is indicated. **e** Mean number of PICH-coated UFBs per anaphase (at least 200 anaphase cells per condition). The significance of differences was assessed in two-tailed unpaired Student's *t*-tests. **f–i, k** Cells were exposed, for 72 h, to aphidicolin (**f**), camptothecin (**g**), hydroxyurea (**h**), and CX-5461 (**i, k**). Each data point is the mean of at least three independent experiments in triplicate. **j** BLM, Tau-5 and CDA protein levels. β-actin was used as a loading control. **l** Immunofluorescence microscopy showing UBTF (*red*) and Tau-1 (*green*) co-localization in BS cells. Merged images (*yellow*) in the bottom panel. *Scale bar*: 5 μm. **m** Schematic representation of a human rDNA repeat (*upper panel*). Chromatin immunoprecipitation (*ChIP*) with Tau-1, UBTF and IgG antibodies in BS cells (*lower panel*). DNA was quantified by qPCR with primer pairs covering the rDNA repeat. Their approximate positions relative to the transcription start site (*TSS*) are indicated on the *x* axis. Data are the means from three independent experiments. **n**, **o** Endogenous pre-rRNA (*45 S*) levels were monitored by qPCR. *B2M*, *β-actin*, and *TBP* were used as housekeeping genes for normalization. The significance of differences was assessed in two-tailed paired Student's *t*-tests. **p**, **q** ChIP with Tau-1 (**p**), UBTF (**q**), and IgG (**p, q**) antibodies. Results are obtained as described in (**m**). **r**, **s** Measurements of pools of NMPs and NTPs. Values of peak surface areas ratios between the endogenous nucleotide and its internal standard are shown. The data are the means of 9 independent measurements corresponding to three independent experiments performed in triplicate. The significance of differences was assessed in Mann–Whitney tests. For IF, DNA was visualized by DAPI staining (*blue* or *white*). *Error bars* represent ± SEM from three independent experiments. ***P < 0.0005, **P < 0.005, *P < 0.05

adjoining rDNA repeat[27, 28]. For confirmation of the specificity of the assay, we performed ChIP-qPCR with UBTF, which is known to interact with the rDNA locus[29, 30]. The endogenous occupancy profile obtained for Tau closely resembled that for UBTF. Tau was specifically associated with the promoter and the transcribed region of the rDNA, but not with the IGS (Fig. 3m). These results,

which were also validated in human embryonic kidney HEK-293T cells (Supplementary Fig. 3j), suggest that Tau may be involved in the regulation of rRNA synthesis. We measured pre-rRNA levels in control cells and in cells with stable or transient Tau depletions. Control qPCR assays on RNA samples were carried out in parallel, in the absence of reverse

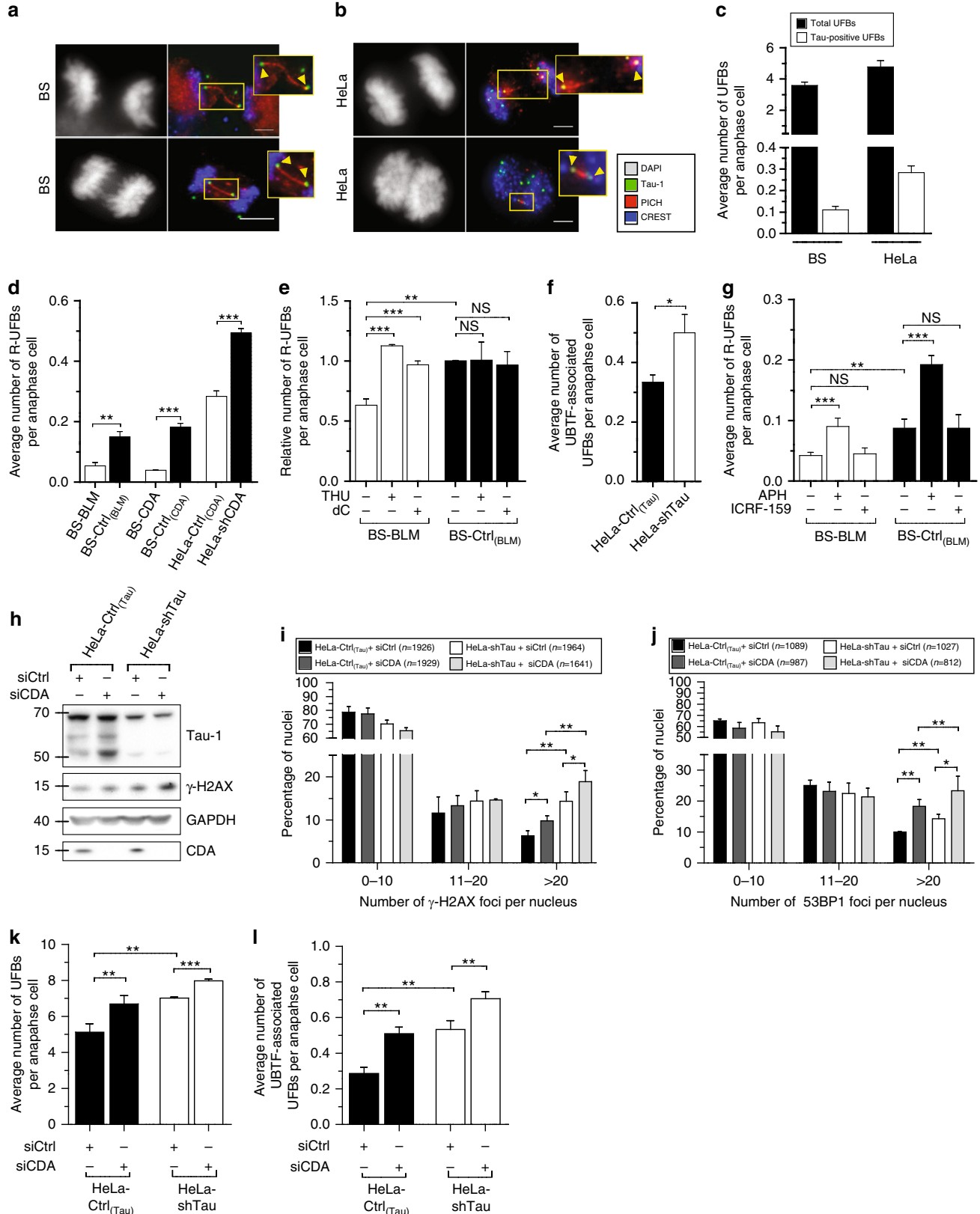

transcriptase, to check that the amplification observed did not result from the presence of contaminating genomic DNA. RT-qPCR analysis revealed that Tau depletion significantly decreased rDNA transcription levels (Fig. 3n). In addition, CDA-deficient cells had higher levels of pre-rRNA synthesis (Fig. 3o), correlated with higher levels of Tau expression (Fig. 3j). Taken together, these data suggest that Tau expression upregulates rRNA synthesis. As Tau expression was correlated with pre-rRNA levels, we investigated the mechanism by which the decrease in Tau levels decreased $45 S$ pre-rRNA synthesis, by performing ChIP-qPCR assays to analyze the effect of Tau downregulation on UBTF recruitment to rDNA repeats. We found that Tau downregulation halved nucleolar Tau binding to rDNA repeats in Tau-depleted cells, as expected (Fig. 3p). Moreover, Tau downregulation also significantly reduced the recruitment of UBTF to the promoter regions of rDNA units (Fig. 3q) without affecting total UBTF protein levels (Supplementary Fig. 3k). Thus, Tau regulates rRNA synthesis through the recruitment of UBTF to rDNA promoters.

Our results showing UBTF and Tau co-localization in nucleolar regions and to rDNA repeats led us to hypothesize that, like Tau deficiency, UBTF deficiency might present a synthetic lethal interaction with CDA deficiency, albeit one not detected in the RNAi screen. We therefore investigated whether UBTF downregulation impaired the survival of CDA-deficient cells. We transfected BS-Ctrl$_{(BLM)}$ and BS-BLM cells with UBTF-targeting siRNAs (Supplementary Fig. 3l). Then, 48 h after transfection, both cell lines were monitored for colony formation. UBTF downregulation had no effect on clonal growth in either cell line (Supplementary Fig. 3m), demonstrating an absence of synthetic lethal interaction between UBTF and BLM and/or CDA deficiencies.

As rRNA synthesis accounts for 35–60% of all transcription[31], we hypothesized that the observed inhibition of pre-rRNA synthesis might be linked to a decreased in the intracellular concentration of ribonucleotides. We therefore evaluated the sizes of the nucleotide pools in parental and Tau-null cells. Liquid chromatography tandem mass spectrometry (LC–MS/MS) analyses showed that Tau knockdown markedly decreased intracellular levels of UMP, GMP, CMP and AMP relative to controls (Fig. 3r and Supplementary Fig. 3n). The UTP, GTP, CTP and ATP pools were also smaller in Tau-deficient cells (Fig. 3s and Supplementary Fig. 3o). However, Tau depletion did not affect intracellular levels of purine and pyrimidine deoxynucleotide monophosphates (dNMPs) and triphosphates (dNTPs), demonstrating that the genetic instability observed in Tau-deficient cells did not result from disequilibrium of the dNMP or dNTP pools (Supplementary Fig. 3p–s). These results demonstrate that ribonucleotide monophosphate and

triphosphate levels are low in Tau-deficient cells, consistent with our finding of a significant decrease in rDNA transcription after Tau depletion (Fig. 3n).

**Tau loss alters genetic integrity of rDNA and CDA$^{-/-}$ cells.** Different types of UFBs have been described in human cells, extending from centromeres (C-UFBs), common fragile sites (CFS-UFBs), or telomeres (T-UFBs)[32–34]. UFBs can be visualized by detection of the helicase-like protein, PICH (Plk1-interaction checkpoint "helicase")[35] which decorates UFBs along their entire length. Other proteins, such as BLM, can also be recruited to bridges but only to pre-existing PICH-coated UFBs[34]. Thus, the total UFB population can be detected only by staining for PICH. Nielsen and Hickson reported that rDNA marks a novel class of UFBs in chicken cells[36], but did not determine whether these DNA structures were also present in human cells. We investigated whether human rDNA loci were prone to ultrafine anaphase bridging, by co-immunostaining rDNA loci and UFBs, with anti-Tau-1 and anti-PICH antibodies, respectively. Given the close proximity of rDNA loci and centromeres on human acrocentric chromosomes, we also labeled centromeres with an antibody against human kinetochore/centromere-specific CREST serum[37] to distinguish between C-UFBs and Tau-positive UFBs. We found that mitotic Tau foci were localized at the termini of UFBs, in a pattern similar to that observed for C-UFBs or CFS-UFBs (Fig. 4a, b and Supplementary Fig. 4a, b). In both cell lines, Tau-positive UFBs were rare, accounting for only ~3–6% of all bridges (Fig. 4c). We showed above that Tau co-localizes with UBTF on the NORs during mitosis, suggesting that Tau-positive UFBs might also display positive staining for UBTF. We thus explored whether UBTF also hung on to the ends of UFBs and evaluated the prevalence of UBTF-positive UFB events. As expected, UBTF-positive UFBs were detected, with a prevalence similar to that of Tau-positive UFBs (Supplementary Fig. 4c, d), indicating that this new subset of UFBs does indeed extend from rDNA clusters facing replication disturbance. These results reveal the existence of a new class of UFBs in human cells, rDNA-associated UFBs, hereafter referred to as R-UFBs.

We investigated the effect of CDA loss on the frequency of R-UFBs. The mean number of R-UFBs per anaphase cell was significantly higher in CDA-defective cells than in control cells (Fig. 4d). We determined R-UFB frequency after the treatment of cells with tetrahydrouridine (THU), a potent CDA inhibitor[38], or with dC, which is known to increase the intracellular concentrations of dC and dCTP[7, 9], to mimic the effects of CDA deficiency. THU or dC treatment had no effect on the prevalence of R-UFBs in CDA-deficient cells, but significantly increased the number of R-UFBs in CDA-proficient cells (Fig. 4e and Supplementary

**Fig. 4** Tau loss alters genetic integrity of rDNA and CDA$^{-/-}$ cells. **a**, **b** Representative immunofluorescence (*IF*) *z*-projection images showing paired Tau foci (*green*) linked by PICH-positive UFBs in BS (**a**) and HeLa (**b**) anaphase cells. *Scale bar*: 5 μm. DNA was visualized by DAPI staining (*white*). Centromeres were stained with CREST serum (*blue*) and UFBs were stained with PICH antibody (*red*). In the enlarged images, Tau foci at the extremities of UFBs are indicated by yellow arrows. **c** Mean number of total and Tau-positive UFBs per anaphase cell. **d**, **f** Mean number of R-UFBs per anaphase cell in untreated cell lines. **e** Relative number of R-UFBs per anaphase cell in BS-Ctrl$_{(BLM)}$ and BS-BLM cells left untreated or treated with 100 μM tetrahydrouridine (*THU*) for 2 × 48 h or 1 mM of deoxycytidine (*dC*) for 10 h. **g** Mean number of R-UFBs per anaphase cell in BS-Ctrl$_{(BLM)}$ and BS-BLM cells left untreated or treated with 0.4 μM aphidicolin (*APH*) or 1 μM ICRF-159 for 24 h. The results were normalized against those for control conditions (mock), which were set to 1. **h** HeLa-Ctrl$_{(Tau)}$ and HeLa-shTau cells were first transfected with either non-targeting or CDA siRNA. After 48 h, cells were again transfected with the indicated siRNAs. Three days after the second round of transfection, cells were collected for western blotting, with GAPDH as the loading control (**h**), or were immunostained with antibody against γ-H2AX or 53BP1 (**i**, **j**). At least 1500 cells were acquired by wide-field microscopy, and γ-H2AX (**i**) or 53BP1 (**j**) foci were counted. The number of cells (*n*) for each condition is indicated. **k**, **l** Cells were immunostained with antibodies against PICH and UBTF, and the mean numbers of total and rDNA-associated UFBs per anaphase cell were monitored. For IF experiments, DNA was counterstained with DAPI. For UFB counting, at least 200 cells were acquired by wide-field microscopy. For all data bars, *error bars* represent ± SD from at least three independent experiments. The significance of differences was assessed in two-tailed unpaired Student's *t*-tests. \*\*\*$P < 0.0005$, \*\*$P < 0.005$, \*$P < 0.05$, *NS* not significant

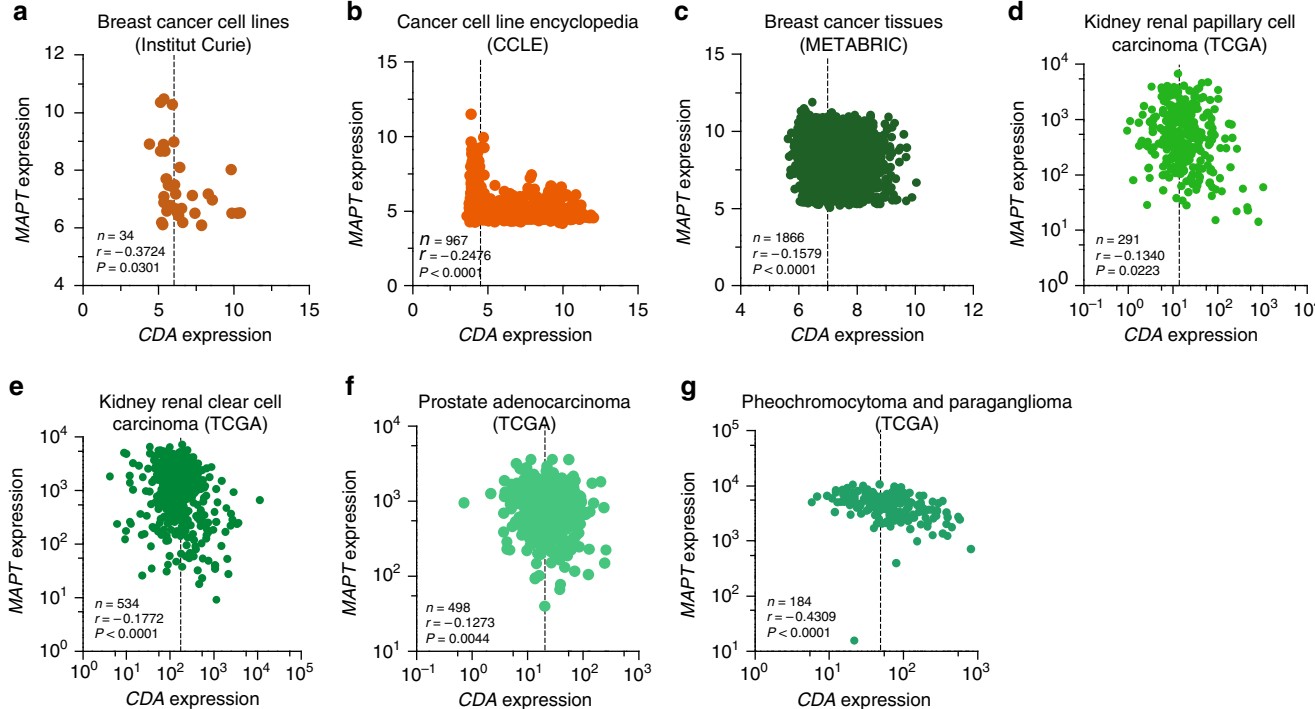

**Fig. 5** *CDA* and *MAPT* levels are negatively correlated in cancer. Scatterplots showing the Pearson correlation between mRNA microarray data **a**–**c** or mRNA sequencing data **d**–**g** for *CDA* and *MAPT* expression for **a** Institut Curie breast cancer cell lines ($n = 34$), **b** CCLE (Cancer Cell Line Encyclopedia) cancer cell lines ($n=967$). **c** METABRIC (*Molecular Taxonomy of Breast Cancer International Consortium*) breast cancer tissues ($n = 1866$), and **d**–**g** TCGA (*The Cancer Genome Atlas*) four different cancer tissues. *Dashed vertical lines* correspond to mean *CDA* expression. $P < 0.05$ were considered statistically significant

Fig. 4e), demonstrating the crucial role of CDA proficiency, and the resulting balanced pyrimidine nucleotide pool, in the prevention of R-UFB formation.

Like CDA deficiency, Tau deficiency favors UFB formation (Fig. 3e). We therefore investigated the possible impact of Tau deficiency on R-UFB frequency. UBTF was used as a marker of rDNA loci in cells lacking Tau. Immunostaining assays showed that Tau-depleted cells had more UBTF-associated UFBs than control cells (Fig. 4f), indicating a role for Tau in preventing R-UFB formation.

Exposure to APH induces the expression of fragile sites in the genome[33, 39]. rDNA clusters have been shown to be sensitive to APH and are thought to behave as fragile sites[40–42]. We therefore investigated whether R-UFB events were affected by exposure to APH. As expected, APH treatment significantly increased the numbers of CFS-associated UFBs in CDA-proficient and CDA-deficient cells, as shown by FANCD2 immunostaining (Supplementary Fig. 4f, g). Similarly, R-UFB frequency was also significantly increased by the APH treatment of these cells (Fig. 4g and Supplementary Fig. 4h). A previous study demonstrated that the treatment of chicken DT-40 cells with an inhibitor of topoisomerase IIα increased the number of R-UFBs[43], but we observed no induction of R-UFBs after ICRF-159 treatment in two independent human cell models (Fig. 4g and Supplementary Fig. 4h). By contrast, the frequency of C-UFBs, as revealed by CREST staining, was significantly higher after this treatment, as previously reported[33] (Supplementary Fig. 4i, j). Thus, the replication stress resulting from CDA depletion or the partial inhibition of replication favors ribosomal ultrafine anaphase bridging.

Finally, we explored how simultaneous depletion of the CDA and Tau proteins led to a synthetic lethal interaction. We therefore monitored γ-H2Ax and 53BP1 foci frequency, and UFB formation after simultaneous Tau and CDA downregulation. As expected, in control cells, the depletion of CDA alone led to increases in γ-H2AX levels, the number of 53BP1 foci, and the prevalence of both total UFBs and R-UFBs (Fig. 4h–l). Moreover, CDA depletion led to an increase in Tau mRNA and protein levels (Fig. 4h and Supplementary Fig. 4k). Finally, down-regulating CDA expression in Tau-deficient cells led to a marked shift towards the presence of large numbers of γ-H2AX and 53BP1 foci in nuclei and an increase in the numbers of total and rDNA-associated UFBs (Fig. 4h–l). Thus, the simultaneous depletion of both proteins aggravated genome instability.

**CDA and MAPT levels are negatively correlated in cancer.** BS cells and CDA-deficient HeLa cells overexpressed Tau, as described above. Tau levels have been reported to increase in certain subtypes of breast cancer[44]. We have recently shown that CDA expression is downregulated in ~60% of cancer cells and tissues[45]. We therefore investigated the possible correlation between CDA loss and Tau overexpression in physiological conditions. We did this by *in silico* analyses of *MAPT* and *CDA* expression levels in several cohorts of cancer cell lines and tissues. An analysis of two cohorts of cancer cell line samples[45, 46] showed that *MAPT* levels were significantly higher in samples with low *CDA* levels, and vice versa (Fig. 5a, b). We then compared *CDA* and *MAPT* mRNA levels in gene expression data sets for different tissue samples[47–49] (Supplementary Data 3–7). The negative correlation between *CDA* and *MAPT* transcript levels was significant in cancers of several tissues, including breast cancer (Fig. 5c), papillary, and clear cell kidney carcinomas (Fig. 5d, e), prostate adenocarcinoma (Fig. 5f), and pheochromocytoma and paraganglioma (Fig. 5g). Overall, our data indicate the existence of a causal link between the expression of the *CDA* and

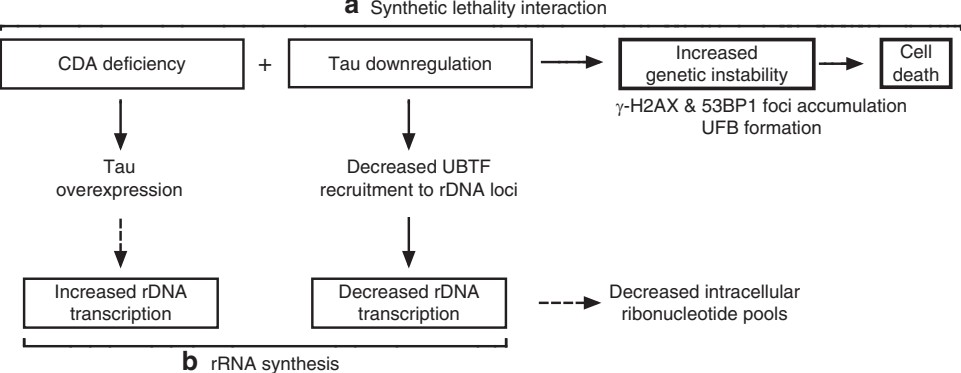

**Fig. 6** Schematic model of the interactions between CDA and Tau. **a** The simultaneous silencing of Tau and CDA increased genetic instability, as illustrated by the increases in γ-H2AX and 53BP1 focus events and UFB frequency. The accumulation of DNA damage and replication stress led to a decrease in clonal growth, and, thus, to cell death. **b** Tau downregulation affected rRNA synthesis by decreasing UBTF recruitment to rDNA promoters, thereby impairing the transcription of rDNA. By contrast, Tau overexpression in CDA-deficient cells led to an increase in rDNA transcription. Finally, Tau downregulation decreased the size of the intracellular ribonucleotide pools, consistent with previous reports linking decreases in rRNA production to imbalances in ribonucleotide pools

*MAPT* genes that could be exploited in the development of new anti-cancer strategies.

## Discussion

In this study, we demonstrate an unexpected role for the microtubule-associated protein Tau in the survival of CDA-deficient cells; the concomitant depletion of both the Tau and CDA proteins promotes genetic instability and is synthetic lethal. H2AX phosphorylation and 53BP1 recruitment events have emerged as a highly specific and sensitive molecular markers for the monitoring of DNA damage initiation and repair[50]. In Tau-deficient HeLa cells, high levels of γ-H2AX and large numbers of 53BP1 foci were observed, indicating an early cellular response to the induction of double strand breaks. This finding is consistent with previous studies showing higher levels of neuronal DNA damage in Tau knockout (KO) mice than in controls. However, γ-H2AX events did not accumulate in Tau KO mice under physiological conditions, and were instead observed after heat stress (HS)-induced damage[11]. This may reflect differences in the DNA damage accumulation in neuronal and non-neuronal cells. UFBs, another marker of genomic instability are considered normal, but their frequency often increases in conditions of constitutive or induced replication stress[33, 34, 51]. Tau depletion increased UFB prevalence in our study. Thus, in addition to suffering DNA damage, Tau-deficient cells undergo endogenous replication challenges. Tau bears some similarity to CDA in that deficiencies of either of these proteins undermine genome integrity. A deficiency of both Tau and CDA resulted in particularly high levels of γ-H2AX and large numbers of 53BP1 foci, and a higher prevalence of UFBs. Thus, the mechanisms underlying synthetic lethality involve impaired DNA repair and replication processes.

Tau may be regarded as a multifunctional protein, the precise role of which depends on its localization. In the cytoskeleton, Tau mediates microtubule polymerization and stabilization[17]. Within the nucleus, there is evidence to suggest that Tau is involved in DNA protection and the promotion of chromosome stability[11, 21, 52]. Nuclear Tau has also been localized to the fibrillary component of the nucleolus and to the NORs of acrocentric chromosomes in human cells[53, 54]. By the double-fluorescence labeling of BS cells with anti-UBTF and anti-Tau-1 antibodies, we confirmed that Tau was localized in the nucleolus. During interphase, these two proteins were distributed as small bead-like structures potentially corresponding to individual transcriptional units. At the end of the G2

phase, when rDNA transcription is switched off[55], UBTF and Tau collect together at mitotic NORs.

Through its ability to bind to the transcribed and control regions of rRNA genes, UBTF has been implicated in PIC formation, transcription initiation, and elongation[56, 57]. The total ablation of UBTF eliminates rRNA gene transcription and leads to changes in rDNA chromatin[58]. As Tau and UBTF were co-localized at nucleolar sites and Tau downregulation impaired the recruitment of UBTF to rDNA promoter regions, it was tempting to speculate that UBTF loss was also synthetic lethal with CDA deficiency. UBTF knockdown seemed to favor DNA damage and genomic instability[59], but our data demonstrated that UBTF downregulation was not synthetic lethal with CDA deficiency.

As nucleolar regions contain rRNA genes, we propose a putative role for Tau in rRNA synthesis. Indeed, we found that the downregulation of Tau expression decreased levels of 45 S pre-RNA, by affecting the recruitment of UBTF to rDNA repeats, consistent with a role for Tau in the regulation of rRNA synthesis. Moreover, Tau depletion markedly decreased the intracellular concentrations of ribonucleotide monophosphates and triphosphates, but not those of deoxyribonucleotides. Cytoskeletal structures have been implicated in the regulation of metabolic enzyme activity and microtubule depolymerization, leading to a strong imbalance in total cell nucleotide pools[60, 61]. The decrease in ribonucleotide levels in Tau-deficient cells may, therefore, result from a loss of the microtubule-associated function of Tau. This decrease in ribonucleotide pool metabolism in Tau-deficient cells may account for their lower levels of rRNA synthesis.

The functional importance of Tau in protecting the integrity of neuronal genomic DNA is directly linked to the capacity of this protein to bind to DNA and form DNA-protein complexes[62, 63]. In this context, we show here, for the first time, that Tau binds throughout the rDNA loci, with a preference for the promoter and transcriptionally active regions. Furthermore, we demonstrate that human rDNA repeats face endogenous replication challenges and form R-UFBs. These results are supported by recent data revealing that a fraction of UFBs in chicken cells are derived from rDNA[43]. The high G-C content of rDNA loci might render them prone to the formation of G-quadruplex structures, causing fork stalling and/or arrest and a slowdown of rDNA replication[64]. Furthermore, we showed that the partial inhibition of replication with APH increased the prevalence of R-UFBs. This finding is consistent with previous reports that exposure to APH

weakens rDNA cluster stability, with these clusters behaving like fragile sites[40–42].

In BS fibroblasts, HeLa cells and breast cancer cell lines and tissues, low levels or absence of CDA were associated with higher levels of Tau mRNA and protein, consistent with a physiological mechanism regulating the levels of both proteins. Based on these data, it is tempting to suggest that Tau overexpression in cells and tissues with little or no CDA is one of the "emergency" systems developed by cells, to protect themselves from the excessive DNA damage and replication stress resulting from CDA loss per se. However, this is not a general rule, as some CDA-low breast cancer cells and tissues did not have high levels of Tau, suggesting that there may be other mechanisms for overcoming excessive DNA damage and replication stress. Nevertheless, a key goal in cancer research is the discovery of new drug targets for selectively impairing the viability of cancer cells. Our finding is, therefore, of major interest. First, several reports have suggested that Tau levels may have prognostic or predictive value in some cancers, due to effects on the efficacy of microtubule-targeted therapies and patient care (reviewed in ref. [44]). Second, CDA is increasingly being seen as important, based on our finding that its deficiency significantly increased sensitivity to aminoflavone treatment[45]. Thus, we suggest that targeting Tau in CDA-deficient cells or targeting CDA in Tau-deficient cells may be a useful option, guiding treatment choices, and increasing the efficacy of targeted therapies.

In conclusion, our results show that Tau depletion impairs the survival of CDA-deficient cells by favoring cellular DNA damage and replication stress. Moreover, Tau depletion affects rRNA synthesis and ribonucleotide pool balance, and undermines the stability of rDNA loci (Fig. 6). Human rDNA loci display greater instability in cancer and aging-related diseases[41, 65], and ribosomal RNA synthesis often increases in cancer cells[66]. Our data, showing an overexpression of Tau in cancer cell lines and tissues with little or no CDA, highlight the importance of analyzing CDA and Tau expression levels simultaneously in cancer tissues, as a relevant and predictive marker of susceptibility to anti-cancer therapies.

## Methods

**Cell lines and drug treatments**. BS GM08505B and HeLa cells were purchased from Coriell Institute and ATCC, respectively. Cell lines were cultured in Dulbecco's modified Eagle's medium (DMEM) supplemented with 10% FCS. BS-Ctrl$_{(BLM)}$ and BS-BLM cells were obtained by transfecting BS GM08505B cells with the EGFP-C1 vector alone (Clontech) or with the same vector containing the full-length BLM cDNA, respectively, using JetPEI reagent. After 48 h, transfectants were selected on 800–1600 µg ml$^{-1}$ G418 (PAA). Individual colonies were isolated and cultured in medium containing 500 µg ml$^{-1}$ G418.

BS-Ctrl$_{(CDA)}$ and BS-CDA cell lines were obtained by transfecting BS-Ctrl$_{(BLM)}$ cells with an empty pCI-puro vector, or with the same vector containing the full-length CDA cDNA (NM001785), using JetPEI reagent. After 48 h, selection was carried out with 0.2 µg ml$^{-1}$ puromycin (PAA) and 500 µg ml$^{-1}$ G418 (Invitrogen). Individual colonies were isolated and maintained in culture with 0.1 µg ml$^{-1}$ puromycin and 500 µg ml$^{-1}$ G418.

HeLa-Ctrl$_{(CDA)}$ and HeLa-shCDA cells were obtained by transfecting cells with an empty pGIPZ vector or with the same vector encoding a short hairpin RNA sequence directed against CDA (Open Biosystems, clone V3LHS_369299), respectively, with JetPEI reagent. After 48 h, transfectants were selected on 1–5 µg ml$^{-1}$ puromycin (Invivogen). Individual colonies were isolated and cultured in medium containing 1 µg ml$^{-1}$ puromycin.

HeLa-Ctrl$_{(Tau)}$ and HeLa-shTau cells were obtained by transfecting cells with an empty pRS vector or with the same vector encoding a short hairpin RNA sequence directed against MAPT (Origene, TR311569), respectively, with JetPEI reagent. After 48 h, transfectants were selected on 1–5 µg ml$^{-1}$ puromycin (Invivogen). Individual colonies were isolated and cultured in DMEM supplemented with 2 µg ml$^{-1}$ puromycin. All cell lines were routinely checked for mycoplasma infection. Authenticity was assessed by comparing the generated short tandem repeat profile with the profiles present in the Deutsche Sammlung von Mikroorganismen und Zellkulturen.

Deoxycytidine (dC, #D5412), aphidicolin (APH, #A0781), ICRF-159 (Razoxane, #R8657), hydroxyurea (HU, #H8627), actinomycin D (ActD, #A9415),

α-amanitin (α-AMA, #A2263), and camptothecin (CPT, #C9911) were purchased from Sigma-Aldrich; THU was purchased from Calbiochem (#584223); CX-5461 (#HY-13323) was purchased from MedChem Express. Drugs were added to the cell culture medium at the following concentrations and for the following amounts of time: dC, 1 mM for 10 h; APH, 0.4 µM for 24 h; ICRF-159, 1 µM for 24 h; THU, 100 µM for 96 h (2 × 48 h). Drugs used for viability assays were added at various concentrations, for 72 h.

**Genome-wide shRNA screen and deep sequencing**. BLM-proficient and BLM-deficient cells were plated in six 150 mm plates, in triplicate (18 plates for each cell line), and transduced at a MOI of 0.8 with pooled lentiviral supernatants from the Open Biosystems GIPZ lentiviral human shRNAmir library (60,000 shRNAs) obtained from the University of Massachusetts Medical School RNAi Core Facility[12]. Each plate was divided into two populations: half of the cells for each replicate were pooled (six plates) and genomic DNA was extracted (referred to as "40 h"), whereas the other half were passaged in the 150 mm plates by four-fold dilutions, as required, for 20 days, at which point the genomic DNA was extracted (referred to as "20 days"). We analyzed the frequency of individual shRNAs in the three replicates of the four populations, using 72 µg of genomic DNA as the substrate (split into 24 tubes) for PCR amplification (94 °C for 1 min, followed by 15 cycles of 94 °C for 1 min, 58 °C for 1 min, 72 °C for 45 s; 72 °C for 10 min) with primers 5′-GAGTTTGTTTGAATGAGGCTTCAGTAC-3′ and 5′-CGCGTCCTAGGTAATACGAC-3′, targeting the core region of GIPZ. The PCR product was gel-purified, and 50 ng of DNA was used as the substrate for a second PCR amplification (94 °C for 1 min; 12 cycles of 94 °C for 1 min, 50 °C for 1 min, 72 °C for 45 s; 72 °C for 10 min) with indexed Illumina primers: forward (5′-CAAGCAGAAGACGGCATACGAGATNNNNNNGTGACTGGAGTTCA-GACGTGTGCTCTTCCGATCTGAGTTTGTTTGAATGAGGCTTCAGTAC-3′) and reverse (5′-AATGATACGGCGACCACCGAGATCTACACTCTTTCCCTA-CACGACGCTCTTCCGATCTTACATCTGTGGCTTCACTA-CATCTGTGGCTTCACT-3′). The PCR product was purified with a Qiagen kit, subjected to quality control and sequenced on the Hiseq Illumina platform. Overall, 678 million sequences aligned with the ENSEMBL v70 transcripts by Bowtie were analyzed by DESeq to identify shRNAs specifically depleted during the growth of the BLM-deficient cells. We rejected shRNAs that were highly toxic to the BLM-complemented cells (decreasing their viability by more than 50%). Hits were selected on the basis of their relative abundance, which was significantly lower 20 days after infection than before transfection in BLM-deficient cells (at least 10-fold) but not in BLM-complemented cells.

**Transcriptomic analysis**. The analysis was performed on BS-Ctrl$_{(BLM)}$ and BS-BLM cells[7]. For each cell line, three independent total RNA samples were extracted using the RNeasy system (Qiagen). Fragmented biotinylated antisense RNA was generated and hybridized to Affymetrix Human Genome U133 Plus 2.0 Arrays. After hybridization, arrays were scanned following guidelines from Affymetrix (http://www.affymetrix.com). These arrays contain ∼54,000 probesets, representing ∼47,000 transcripts. Data were input into R and normalized with the robust multiarray average expression measure[67]. Differential expression was then assessed with a linear model and the Bioconductor limma package[68]. Differentially expressed genes were identified on the basis of a |fold change| ≥ 1.5 and $P < 0.05$.

**Transfection with siRNA and colony formation assay**. Cells were transfected with a pool of four siRNAs specific for MAPT, UBTF, CDA or BLM (ON-TAR-GETplus SMART-pool, Dharmacon) or negative control siRNAs (ON-TARGET-plus siCONTROL Non Targeting Pool, Dharmacon), in the presence of DharmaFECT 1 (Dharmacon). We used a standard siRNA concentration of 50 nM. The sequences of the siRNAs are provided in Supplementary Table 6. The colony formation assays were carried out in 6-well or 12-well plates. Cells were counted and used to seed plates in a serial dilution series, in at least triplicate. After 7 and 10 days of growth for HeLa and BS cells, colonies were fixed and stained with crystal violet. Colonies were scored with ImageJ. Only experiments giving a linear correlation between the different dilutions were considered. Colony-forming efficiency was estimated by dividing the number of colony-forming units by the number of cells plated.

**Immunoblotting**. Proteins were isolated in lysis buffer (8 M urea, 50 mM Tris-HCl, pH 7.5, and 150 mM β-mercaptoethanol), separated by electrophoresis in 4–12% Bis-Tris pre-cast gels (NuPAGE Novex, Life Technologies), and blotted onto polyvinylidene fluoride membranes, which were then incubated with appropriate primary and secondary antibodies. Bands were visualized by chemiluminescence (Clarity Western ECL Substrate, Bio-Rad) with a ChemiDoc XRS+Molecular Imager and Image Lab software. Uncropped immunoblots are presented in Supplementary Fig. 5. The following antibodies were used for detection: mouse anti-Tau-5 (#ab80579, Abcam, dilution 1:1000), mouse anti-Tau-1 (clone PC1C6, #MAB3420, Millipore, dilution 1:1000), rabbit anti-CDA (#ab56053, Abcam, dilution 1:500), rabbit anti-BLM (#ab2179, Abcam, dilution 1:5000), rabbit anti-UBTF (#sc-9131, H-300, Santa Cruz, dilution 1:500), rabbit anti-γH2AX (S139) (#2577, Cell Signaling, dilution 1:500), mouse anti-GAPDH (#G8795, Sigma-Aldrich, dilution 1:10,000), rabbit anti-β-actin (#A2066, Sigma-Aldrich,

dilution 1:10,000), horseradish peroxidase (HRP)-conjugated goat anti-rabbit IgG (#sc-2054, Santa Cruz, dilution 1:5000), and HRP-conjugated goat anti-mouse IgG (#sc-2055, Santa Cruz, dilution 1:5000).

**Reverse transcription and real-time quantitative PCR.** Total RNA was extracted with the NucleoSpin RNA kit (Macherey-Nagel). RNA quality was assessed with a NanoDrop 2000 spectrophotometer, and cDNAs were synthesized with the iScript Advanced cDNA synthesis kit (Bio-Rad) and 1 µg of RNA. Real-time PCR was performed with the cDNA template (1/10 dilution), iQ SYBR Green Supermix (Bio-Rad), and 300 nM forward and reverse primers. Amplification was performed with the CFX96 detection system (Bio-Rad). The relative quantities of the MAPT, CDA and BLM cDNAs were normalized against three reference genes (B2M, β-actin and TBP). The primer sequences are provided in Supplementary Table 6.

**Antibodies and immunofluorescence microscopy.** The primary antibodies and dilutions used were: mouse anti-Tau-1 (PC1C6, #MAB3420, Millipore, dilution 1:500); rabbit anti-Tau (phospho Thr231) (#ab195739, Abcam, dilution 1:500); rabbit anti-Tau (phosphor Ser396) (#ab109390, Abcam, dilution 1:1000); rabbit anti-UBTF (H-300, #sc-9131, Santa Cruz, dilution 1:100); mouse anti-UBTF (F-9, #sc-13125, Santa Cruz, dilution 1:100); rabbit anti-PICH (#H00054821-D01P, Abnova, dilution 1:150); mouse anti-PICH (3F12-2B10, #H00054821-M01, Abnova, dilution 1:100); rabbit anti-FANCD2 (#NB100-182, Novus Biologicals, dilution 1:200); human anti-CREST (#15-234-0001, Antibodies Online GmbH, dilution 1:100); rabbit anti-γH2AX (#39117, Active Motif, dilution 1:500); mouse anti-53BP1 (clone BP13, #MAB3802, Millipore, dilution 1:500). The secondary antibodies and dilutions used were: goat anti-rabbit Alexa Fluor 555, 1:500; goat anti-mouse Alexa Fluor 488, 1:500; goat anti-human Alexa Fluor 633, 1:500; goat anti-rabbit Alexa Fluor 488, 1:500. For Tau-1, PICH, CREST, UBTF and FANCD2 immunostainings, cells grown on coverslips were fixed with 4% paraformaldehyde, permeabilized with 0.5% Triton X-100 and blocked by incubation with 5% bovine serum albumin. For γH2AX and 53BP1 immunostaining, cells grown on coverslips were first treated with cytoskeletal buffer (CSK, 100 mM NaCl, 300 mM sucrose, 3 mM MgCl$_2$, 10 mM Hepes pH 6.7, and 0.5% Triton X-100), and then fixed with 4% paraformaldehyde, permeabilized with 0.2% Triton X-100 and blocked by incubation with 5% bovine serum albumin/Tween 0.1%. Immunofluorescence staining was carried out by incubating with primary antibodies, followed by secondary antibodies. Coverslips were mounted on glass slides in Prolong Gold antifade reagent containing DAPI (Life Technologies). Images were captured with a 3-D imaging system consisting of a Leica DM RXA microscope equipped with a piezoelectric translator (PIFOC; PI) using a×100 or ×63 objective, and were collected as a stack of 0.2 µm increments on the z-axis (Metamorph software; Molecular Devices). ND_Safir software was used for noise reduction[69]. Images are presented as maximum intensity projections, generated with ImageJ software, from stacks deconvolved with an extension of Metamorph software. γ-H2AX and 53BP1 foci per nucleus were counted by a customized macro using a semi-automated procedure, as follows: the nucleus stack was first smoothed using a median filter (radius 5), the user defined an intensity value as a threshold (one value for all experiments); a mask was then generated and transferred onto the stack of foci so that only foci in nuclei were analyzed. A top-hat filter was applied to the result to eliminate local background and facilitate the segmentation process, based on a simple threshold (user defined value). Finally, the macro counted and characterized foci. At least 400 nuclei were analyzed for each condition.

**Chromatin immunoprecipitation.** Cells were cross-linked by incubation with 0.75% formaldehyde at room temperature for 10 min, with quenching in glycine to a final concentration of 0.125 M for 5 min. Cells were then collected, washed twice with ice-cold 1×phosphate-buffered saline, and resuspended in RIPA buffer. Cells were incubated on ice for 10 min and chromatin was sonicated with a Bioruptor Pico (Diagenode) for 10 cycles (BS and HEK-293T cells), or 20 cycles (HeLa cells), to obtain an average DNA fragment size of 200–1000 bp. The resulting lysate was precleared by centrifugation for 10 min at 4 °C and stored at −80 °C for immunoprecipitation. We cross-linked 10 µg of anti-Tau-1, 3 µg of anti-UBTF, and 5 µg of anti-IgG antibodies to the beads (Dynabeads Protein G) by incubation with dimethyl pimelimidate (DMP, 13 mg ml$^{-1}$) for 30 min, and then incubated the beads with chromatin lysate overnight at 4 °C, with rotation. The beads were sequentially washed with 500 µl each of the following buffers: low-salt wash buffer (0.1% SDS, 1% Triton X-100, 2 mM EDTA, 20 mM Tris-HCl pH 8, 150 mM NaCl), high-salt wash buffer (0.1% SDS, 1% Triton X-100, 2 mM EDTA, 20 mM Tris-HCl pH 8, 500 mM NaCl), and LiCl wash buffer (0.25 mM LiCl, 1% Nonidet P-40, 1% sodium deoxycholate, 1 mM EDTA, 10 mM Tris-HCl pH 8). The immune complexes were eluted with 150 µl of freshly prepared elution buffer (1% SDS, 100 mM NaHCO$_3$). The cross-linking reaction was reversed by overnight incubation of the eluents with 0.2 M NaCl at 65 °C, and the DNA was then treated with 1 µg of each RNase A and proteinase K, and recovered with a PCR purification kit (Qiagen). Approximately 5% of the bound DNA fraction was used for quantitative PCR. All primers used in qPCR analyses are detailed in Supplementary Table 6. Data were normalized against non-specific genomic DNA and relative to IgG.

**Cell viability assay.** Cells were plated at a density of $2\times10^3$ per well in 96-well microplates, in triplicate. After 24 h, cells were left untreated or treated with various doses of each drug. After 72 h, cell viability was assessed with 3-(4,5-dimethyl-2-thiazolyl)-2,5 diphenyl-2H-tetrazolium bromide (MTT; Life Technologies).

**Nucleotide determinations.** The determination of ribonucleotide triphosphates (NTP: ATP, UTP, GTP, and CTP), deoxyribonucleotide triphosphates (dNTP: dATP, dCTP, dGTP, and dTTP), ribonucleotide monophosphates (NMP: AMP, UMP, GMP, and CMP), and deoxyribonucleotide monophosphates (dNMP: dAMP, dCMP, dGMP, and dTMP) was performed with a published method based on an online extraction coupled with LC–MS/MS[70]. Briefly, online extraction was performed with an Oasis-WAX column (3.9 × 20 mm; 30 µm—Waters), followed by a Hypercarb analytical column (2.1 × 100 mm; 5 µm—ThermoScientific) for the separation of the compounds. Gradient elution was performed with a constant proportion of NH$_4$OH (0.25%—pH 10), varying proportions of water and acetonitrile. An electrospray ionization source was used and the positive mode was selected for detection of the compounds. For each ribonucleotide and deoxyribonucleotide, the peak surface area ratio was determined between the endogenous nucleotide and the corresponding labeled nucleotide used as an internal standard (adenosine-$^{13}$C$_{10}$,$^{15}$N$_5$ 5′-triphosphate (ATP$_{13C,15N}$), uridine-$^{13}$C$_9$,$^{15}$N$_2$ 5′-triphosphate (UTP$_{13C,15N}$), thymidine-$^{13}$C$_{10}$,$^{15}$N$_2$ 5′-triphosphate (TTP$_{13C,15N}$), cytidine-$^{15}$N$_3$ 5′-triphosphate (CTP$_{15N}$), guanosine-$^{13}$C$_{10}$ 5′-triphosphate (GTP$_{13C}$), adenosine-$^{13}$C$_{10}$,$^{15}$N$_5$ 5′-monophosphate (AMP$_{13C,15N}$), thymidine-$^{13}$C$_{10}$,$^{15}$N$_2$ 5′-monophosphate (TMP$_{13C,15N}$), uridine-$^{15}$N$_2$ 5′-monophosphate (UMP$_{15N}$), cytidine-$^{15}$N$_3$ 5′-monophosphate (CMP$_{15N}$), guanosine-$^{13}$C$_{10}$,$^{15}$N$_5$ 5′-monophosphate (GMP$_{13C,15N}$), 2′-deoxyadenosine-$^{13}$C$_{10}$,$^{15}$N$_5$ 5′-triphosphate (dATP$_{13C,15N}$), 2′-deoxycytidine-$^{13}$C$_9$,$^{15}$N$_3$ 5′-triphosphate (dCTP$_{13C,15N}$), 2′-deoxyguanosine-$^{13}$C$_{10}$,$^{15}$N$_5$ 5′-triphosphate (dGTP$_{13C,15N}$), 2′-deoxyadenosine-$^{13}$C$_{10}$,$^{15}$N$_5$ 5′-monophosphate (dAMP$_{13C,15N}$), 2′-deoxycytidine-$^{13}$C$_9$,$^{15}$N$_3$ 5′-monophosphate (dCMP$_{13C,15N}$), 2′-deoxyguanosine-$^{13}$C$_{10}$,$^{15}$N$_5$ 5′-monophosphate (dGMP$_{13C,15N}$)) making it possible to take into account the matrix effect for each compound. Cell extracts were prepared as follows: the cells were washed with phosphate-buffered saline and subjected to extraction in 3 ml of a mixture of methanol and water (70/30; v/v). The resulting extract was transferred to a tube, shaken for 5 min, and stored at −80 °C until analysis. On the day of analysis, a mixture containing all the internal standards was added to each sample, and the samples were then vortexed and centrifuged for 10 min at 13,000 × g. The supernatant was then evaporated off under nitrogen at 37 °C and the dry residue was resuspended just before injection into the LC–MS/MS apparatus.

**Statistics.** At least three independent experiments were carried out to generate each data set. The statistical significance of differences in colony formation and mRNA levels was determined in two-tailed paired t-tests. Differences in ultrafine anaphase bridge frequencies were assessed in two-tailed unpaired t-tests. Pearson correlation analysis was used to assess the association between CDA and MAPT transcript levels. P-values for nucleotide pool measurements were obtained in Mann–Whitney tests, with values of $P < 0.05$ considered significant.

**Data availability.** Raw and normalized sequencing (synthetic lethality screen) and transcriptomic data are available from the Institut Curie microarray dataset repository (http://microarrays.curie.fr/; Direct links: http://microarrays.curie.fr/publications/ UMR3348/shRNA-based-bloom-syndrome-cells/ and http://microarrays.curie.fr/publications /UMR3348/pyrimidine-bloom-syndrome/). MAPT and CDA expression levels were extracted from a collection of 34 human breast tumor cell lines (mostly from ATCC) established at the Translational Research Department of Institut Curie[45], from the Gene Expression Omnibus (GEO) dataset no. GSE54502[22], from the Cancer Cell Line Encyclopedia (CCLE)[46], from the European Genome-Phenome Archive dataset no. EGAS00000000083[47] and from the TCGA portal[55, 56].

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

## Acknowledgements

We thank David Gentien, Caroline Hégo, Philippe Huppé, Pierre Gestraud, and Philippe La Rosa for helping us with transcriptomic data analysis and online data deposition. We thank the PICT-IBISA imaging facility staff for assistance. We thank Dr. Amy Virbasius for helping to produce the pooled shRNA library. We thank the TCGA network for making available the high-throughput data. We thank Hamza Mameri for stimulating discussions. This work was supported by the Institut Curie (Translational Cancer Research grant and PICSysBio), CNRS, Ligue Nationale Contre le Cancer (Comité de l'Essonne), the Agence Nationale de la Recherche and the Institut National du Cancer. E. B.S. was supported by grants from Institut Curie and Ligue Nationale Contre le Cancer. RNAi screen was supported by grant to X.V. from Ligue Nationale Contre le Cancer (grant number LNCC92 no. MC2012-031). C.G. was supported by CNRS, ATIGE-Genopole, CEA-DSV and an Emergence grant from Canceropole IdF. M.R.G. is a Howard Hughes Medical Institute investigator.

## Author contributions

X.V., C.G., and M.A.G. conceived and initiated the RNAi screen project. E.B.S. and M.A.G. conceived and designed all the other experiments. C.G., M.R.G., and O.A. designed, performed and analyzed the primary RNAi screen. E.B.S., G.B.L and R.O.D. carried out cellular and molecular biology experiments, and analyzed the data. E.B.S. performed and analyzed microscopy experiments. C.M., J.G., and E.B.S. performed nucleotide determinations and analyzed data. E.B.S. and M.A.G. prepared the manuscript. M.A.G. supervised the study.

## Additional information

**Competing interests:** The authors declare no competing financial interests.

