## [Peer Review File · Nature Communications]

Reviewers' Comments:

Reviewer #1 (Remarks to the Author)

In the manuscript "New functions for Tau protein in maintaining ribosomal DNA stability and cytidine deaminase-deficient cell survival" the authors identified Tau as an essential factor to enable BLM- and /or CDA-deficient cells to survive despite persistent DNA damage and replication stress, though its role in rDNA metabolism.

The topic is highly interesting unfortunately a large part of the conclusions are not yet supported by the results.

In all immunoblotting analyses, Tau has been detected using Tau1 antibody. Obviously there was a misunderstanding concerning this antibody. Tau1 is a phospho-dependent antibody that recognizes a non phosphorylated Tau epitope (unP-Ser195-202) and is not suitable to detect a global amount of Tau.

Therefore, in this manuscript, all the quantifications of Tau immunoblotting do not reflect a global level of Tau but a relative level of unphosphorylated Tau at this epitope. Besides, the level of phosphorylation of Tau at epitope Ser195-202 could be analyzed using the ratio Tau1 over total Tau (Tau1/GAPDH over total Tau/GAPDH).

A new set of immunoblotting analyses using a phospho-independent anti-total Tau antibody is mandatory before to conclude about Tau levels throughout this study.

In immunofluorescence analyses, Tau1 is also the only antibody used to visualize Tau in interphase and mitotic cells. Although Tau1 efficiently labels nucleolar Tau in human interphase cells, various studies reported that Tau is highly phosphorylated during mitosis. Labelling using anti-phospho Tau antibodies previously described in mitotic cells would add interesting informations. Anyway this point should be discussed in the manuscript.

Immunofluorescent images have been captured using a 3-D deconvolution system. Show the initial images before applying 3-D deconvolution.

Although the authors claim that γ H2AX is a highly specific marker of DNA damage and repair, a growing body of evidence demonstrates that γ H2AX plays others non canonical roles in cells. Notably an increase of γ H2AX has been reported during mitosis. Therefore, to use another marker, such as 53BP1 is necessary to confirm the presence of DSB DNA damage/repair (Fig. 3a,b). Show immunofluorescent labelings for γ H2AX and 53BP1.

Fig.3j: Run a RT- control to ensure that no genomic DNA contamination occurred.

Reviewer #2 (Remarks to the Author)

Title: New functions for Tau protein in maintaining ribosomal DNA stability and cytidine deaminase-deficient cell survival

Authors: Samra et al.

In this manuscript by Samra et al., the authors have tried to understand the mechanism(s) which allow the survival of cells which lack BLM and/or CDA. Using a range of cell biology based experiments, the authors indicate that a synthetic lethal interaction existed between CDA and Tau in cells which lack BLM and/or CDA. Presence of Tau was necessary for the ribosomal DNA (rDNA) stability. Hence absence of Tau led to the formation of a new class of ultrafine anaphase bridges, which contributed to genome instability along with CDA. Finally the authors postulate a role of Tau in ribosomal RNA synthesis.

The manuscript is novel, generally well written with defined controls. The genetic screen carried out by the authors is appreciated by this reviewer. However the cell biology experiments done to test the hypothesis require additional experiments and validation, before the manuscript can be considered for publication in Nature Communications.

Major critiques:

1. The authors need to find a defined mechanism by which Tau affects ribosomal DNA stability. While the data at this stage strongly suggests that Tau affects indeed genetic/genomic stability, it does not provide compelling evidence in favor of the mechanism(s) necessary for the process. For example, how does Tau regulate rDNA transcription?
2. The authors again do a very thorough job establishing the parameters required for the formation of R-UFB. However there are some inherent contradiction which need to be addressed. If the lack of CDA increases Tau (as the authors have repeatedly shown) – then what is the physiological basic of concomitant silencing of both Tau and CDA?
3. This reviewer is of the viewpoint that the in vivo link between the transcript levels of CDA and MAPT (Tau) in patient cell lines and tissues is very interesting. This should be the final figure of the paper. Additionally the authors should consider supplementing the in vivo data with mouse xenograft experiments using HeLa stable lines overexpressing and/or ablating BLM, Tau and CDA, individually and in combination.
4. Figure 1E, 2A, 2D – the decrease in the levels of Tau is consistently quantitated in Figure 2B, 2E to around 50%. However the blots do not seem to reflect this. Blots should be redone with lesser amount of lysates or lower exposures provided. Why are SD/SEM not shown in Figure 2B, 2E?
5. Result (Line 160, 161) – to a level similar to that obtained by BLM complementation – can such a statement be made for different blots with possibly different exposures?
6. The authors should provide an explanation why CDA and UBTF were not detected in the screen (Figure 1A).
7. The authors repeatedly show that decrease in the level of CDA causes an increase in the levels of Tau? What is mechanism for this to happen?
8. Figure 3D-3G – the drug sensitivity assays should be done with two other isogenic lines HeLa shBLM and HeLa shCDA and data for all four lines be presented together.
9. Figure 3I – to possibly determine whether Tau enhances the binding of UBTF to the rDNA (see comment 1 above), the authors should try to do a reciprocal re-CHIP using UBTF and CDA antibodies.
10. The authors should try to co-localize Tau with UBTF at the termini of R-UFB. This alone (and not co-staining with either CREST and/or PICH_ should be a criteria to define R-UFBs.
11. A schematic model which indicates the inter-relationship and genetic interactions involving BLM, CDA and Tau needs be to shown.

Other critiques:

1. Abstract, line 28 – It should be Tau is overexpressed in CDA/BLM deficient cells.
2. Since the role of UBTF is quite important in the context of the paper, it should be mentioned in

the abstract.

3. Introduction (line 69-70); The BS cells are susceptible to increase in cell lethality due to depletion....
4. Introduction (line 76) – it is not co-localization, it is co-recruitment.
5. Introduction (line 78, 79) – Tau bound (not interacted) with active rDNA repeats. However at present there is no evidence to suggest that this recruitment/binding favoured rDNA transcription.
6. Figure 3C – which UFBs are the authors counting here?
7. Figure 3H- The nucleolus is not very well defined in the DAPI image. So either a marker for nucleolus be co-stained or alternate photos given.
8. Figure 3L, M – the LC-MS/MS spectra should be put in as supplemental data.
9. Line 292 – the Figure mentioned as 3D, is actually 5D.
10. Line 318, 321 – Figure 6C-6C is actually 6A-6D.

Reviewer #1 (Remarks to the Author):

In the manuscript "New functions for Tau protein in maintaining ribosomal DNA stability and cytidine deaminase-deficient cell survival" the authors identified Tau as an essential factor to enable BLM- and /or CDA-deficient cells to survive despite persistent DNA damage and replication stress, though its role in rDNA metabolism.

The topic is highly interesting unfortunately a large part of the conclusions are not yet supported by the results.

1. In all immunoblotting analyses, Tau has been detected using Tau1 antibody. Obviously there was a misunderstanding concerning this antibody. Tau1 is a phospho-dependent antibody that recognizes a non-phosphorylated Tau epitope (unP-Ser195-202) and is not suitable to detect a global amount of Tau. Therefore, in this manuscript, all the quantifications of Tau immunoblotting do not reflect a global level of Tau but a relative level of unphosphorylated Tau at this epitope. Besides, the level of phosphorylation of Tau at epitope Ser195-202 could be analyzed using the ratio Tau1 over total Tau (Tau1/GAPDH over total Tau/GAPDH). A new set of immunoblotting analyses using a phospho-independent anti-total Tau antibody is mandatory before to conclude about Tau levels throughout this study.

We agree with the reviewer's comment and we apologize for not clarifying this important point in our initial manuscript. We are fully aware of that Tau-1 recognizes unphosphorylated Tau epitope Ser195-202, and that the levels obtained with this antibody do not necessarily reflect global levels of Tau protein, although we found that *MAPT* mRNA levels were perfectly correlated with Tau-1 protein levels (Fig. 2a, 2l and Supplementary Fig. 2c of the

revised version of the manuscript, compared to Fig. 2b, Supplementary Fig. 2h and Supplementary Fig. 2d, respectively).

We have addressed the reviewer's concern and clarified this point, by testing the two antibodies known to recognize total Tau regardless of phosphorylation state, T46 and Tau-5. We were unable to detect Tau in non-neuronal cells (BS and HeLa cells) with the T46 antibody. However, Tau-5 detected Tau in our cell lines, albeit with some difficulty in BS cells, as described below. We therefore performed additional immunoblotting analyses with the Tau-5 antibody. Tau-5 reacts with all human Tau isoforms because it recognizes a sequence in the middle of the region extending between amino acids 210 and 241, regardless of Tau phosphorylation state. It was not easy to detect Tau with the Tau-5 antibody in BS cells, which have much lower levels of Tau than neuron extracts, because the Tau signal was masked by non-specific labeling (Fig. 1a-b for the reviewer only). After detailed technical adjustments and protocol modification, such as prolonged overnight incubation with the Tau-5 antibody, chemiluminescence with prolonged exposure and membrane slicing, we were able to detect Tau proteins. As shown in Fig. 1a-b, for the reviewer only, the non-specific Tau bands were readily detectable after exposure for 10 seconds (Fig. 1a, for the reviewer only). However, Tau-specific bands were detectable only after membrane slicing and chemiluminescence with long periods of exposure (Fig. 1b, for the reviewer only). Our findings confirm that Tau is more strongly expressed in BS-Ctrl_(BLM) cells than in BS-BLM cells. We would like to stress that once a membrane is probed with Tau-5 antibody, it is impossible to probe it again with Tau-1 antibody. This explains why different blots are represented every time for Tau-5 and Tau-1.

Figure 1 | for the reviewer only: BLM and Tau protein quantification in BS-Ctrl_(BLM) and BS-BLM cells. The membrane slices were incubated overnight with primary antibodies and bands were visualized by chemiluminescence with exposure for 10 seconds or 10 minutes.

Despite the lower sensitivity and stronger background obtained with the Tau-5 antibody, our experiments with this antibody confirmed all the results obtained with Tau-1 presented in the manuscript.

- CDA expression, in response to the stable expression of exogenous BLM or CDA, reduced Tau protein levels (Fig. 2b, 2f and 2i of the revised version of the manuscript),
- Tau expression was upregulated in cells depleted of CDA (Fig. 2n and Fig. 3j of the revised version of the manuscript).

These results indicate that global Tau levels are higher in CDA-deficient cells than in control cells, regardless of Tau phosphorylation state.

Thus, in response to the reviewer's comment, we present the results obtained with the Tau-5 and Tau-1 antibodies in the following figures:

- In Fig. 2b and 2c, immunoblots with Tau-5 and Tau-1 antibodies are presented as principal figures.

- In Fig. 2f, 2i and 2m, immunoblots with Tau-5 are presented as the principal figures, whereas the initial immunoblots obtained with Tau-1 now appear as supplementary figures, with high and low levels of exposure for the Tau-1 blots, as requested by reviewer 2 (point 4) (Supplementary Fig. 2d, 2f and 2k).

For the sake of clarity, the paragraph on pages 6-7 lines 138-148 has been shifted from the “*A genome-wide shRNA-based synthetic interaction screen in BS cells*” section of the *Results* to the “*Tau silencing reduces the survival of CDA-deficient cells*” section and has been modified as follows:

“In our transcriptomic data, Tau mRNA levels in BS cells were found to be twice those in controls ($P = 4.78 \times 10^{-2}$, Supplementary Table 4a). By examining microarray expression data for BLM-depleted or nonspecific shRNA-treated isogenic control fibroblasts available under accession no. GSE54502²⁶, we also found that Tau was significantly upregulated in BLM-depleted cells relative to control cells (fold-change = 1.45, $P = 7.63 \times 10^{-4}$, **Supplementary Fig. 2a-b**), supporting our findings. **The overexpression of Tau in BS cells was confirmed by reverse transcription-quantitative PCR (RT-qPCR) and western blotting (WB) (Fig. 2a-c). We used the Tau-1 and Tau-5 monoclonal antibodies, which detect the unphosphorylated Tau epitope Ser195-202, and total Tau regardless of Tau phosphorylation state, respectively (Fig. 2b-c). BLM expression reduced Tau mRNA levels by 59% (Fig. 2a), and the amounts of protein detected by Tau-5 and Tau-1 by 28% and 36.5%, respectively (Fig. 2d-e).**”

Finally, it was difficult for us to analyze the level of Tau phosphorylation at epitope Ser195-202, because Tau-5 was less sensitive than Tau-1 for Tau detection, and gave higher levels of background.

2. In immunofluorescence analyses, Tau1 is also the only antibody used to visualize Tau in interphase and mitotic cells. Although Tau1 efficiently labels nucleolar Tau in human interphase cells, various studies reported that Tau is highly phosphorylated during mitosis. Labelling using anti-phospho Tau antibodies previously described in mitotic cells would add interesting information. Anyway this point should be discussed in the manuscript.

We agree with the reviewer's comment that the use of anti-phospho-Tau antibodies would add interesting information. However, our findings, revealing a greater sensitivity to the RNA polymerase I inhibitor CX-5461 in Tau-depleted cells (Fig. 3i in the revised version of the manuscript) led us to focus on the nucleolar Tau fraction. We therefore chose to use Tau-1, because this is, to our knowledge, the only monoclonal antibody for Tau for which nucleolar staining has been reported. Indeed, co-incubation of the Tau-1 antibody with a synthetic peptide encompassing the Tau-1 epitope inhibited nucleolar Tau-1 staining, clearly demonstrating the presence of the Tau-1 epitope in the nucleolus (Loomis *et al.*, 1990).

However, we have addressed the reviewer's comment, by monitoring the phosphorylation state of Tau in BS cells at individual stages of the cell cycle. We examined interphase and mitotic BS cells by immunofluorescence microscopy with two phosphorylation-dependent antibodies sensitive to phosphorylation of the Thr231 and Ser396 epitopes. The Thr231 and Ser396 residues, along with other Thr-Pro and Ser-Pro motifs, have been reported to be preferentially phosphorylated during mitosis (Preuss *et al.*, 1995; Preuss & Mandelkow,

1998). The cells were fixed with paraformaldehyde in the same way as for Tau-1 immunostaining. As shown in Supplementary Fig. 3h-i of the revised version of the manuscript, very low levels of Tau phosphorylation were detected in interphase cells, whereas Tau became highly phosphorylated during mitosis. In mitotic cells, immunoreactivity with the P-Thr231 antibody was observed at the centrosomes during prometaphase. This immunoreactivity decreased during metaphase and anaphase, and was then observed again in the midbody region during telophase (Supplementary Fig. 3h; Preuss & Mandelkow, 1998). Immunoreactivity with the P-S396 antibody was also localized to the midbody region of anaphase and telophase cells (Supplementary Fig. 3i). These data clearly are consistent with previous studies showing that the very low levels of phosphorylated Tau detected in interphase cells are not present in the nucleolus, and that Tau is highly phosphorylated during mitosis (Preuss *et al.*, 1995; Preuss & Mandelkow, 1998).

On the basis of these results, we have modified the *Results* section pages 9-10 lines 218-232 as follows:

“The Tau-1 antibody efficiently labels nucleolar mitotic Tau through the detection of an unphosphorylated Tau epitope, but it is widely accepted that Tau phosphorylation increases during mitosis in neuronal and non-neuronal cells^{31,32}. We therefore examined interphase and mitotic BS cells by immunofluorescence microscopy with two phosphorylation-dependent antibodies sensitive to phosphorylation of the Thr231 and Ser396 epitopes. As previously reported, very low levels of Tau phosphorylation were detected in interphase BS cells, whereas Tau became highly phosphorylated during mitosis (Supplementary Fig. 3h-i). In mitotic cells, immunoreactivity with the P-Thr231 antibody was observed at centrosomes during prometaphase. This immunoreactivity decreased during metaphase and anaphase, and

was then observed again in the midbody region during telophase (Supplementary Fig. 3h). Immunoreactivity with the P-S396 antibody was also localized to the midbody region in anaphase and telophase cells (Supplementary Fig. 3i). Thus, in BS cells, Tau displays low levels of phosphorylation in interphase cells, but becomes highly phosphorylated during mitosis, as reported in neuronal cells³². However, none of the tested phosphorylation-dependent antibodies localized to the nucleolus or to the NORs.”

3. Immunofluorescent images have been captured using a 3-D deconvolution system. Show the initial images before applying 3-D deconvolution.

The initial images obtained before 3-D deconvolution are provided below, in Fig. 2a-d for the reviewer only:

Figure 2 | for the reviewer only: (a) Immunofluorescence microscopy showing UBTF (red) and Tau-1 (green) colocalization in Bloom's syndrome (BS) cells. DNA was visualized by DAPI staining (white). Merged images (yellow) of Tau-1 with UBTF are shown in the bottom panel. (b-d) Representative immunofluorescence z-projection images showing paired Tau foci (b-c, green) or paired UBTF foci (d, green) linked by PICH-positive UFBs in BS and HeLa anaphase cells. Centromeres were stained with CREST serum (blue) and UFBs were stained with PICH antibody (red). In the enlarged images, Tau foci at the extremities of UFBs are indicated by yellow arrows. DNA was visualized by DAPI staining (white). Scale bar: 5 μ m.

4. Although the authors claim that γ H2AX is a highly specific marker of DNA damage and repair, a growing body of evidence demonstrates that γ H2AX plays others non canonical roles in cells. Notably an increase of γ H2AX has been reported during mitosis. Therefore, to use

another marker, such as 53BP1 is necessary to confirm the presence of DSB DNA damage/repair (Fig. 3a,b). Show immunofluorescent labeling for γ H2AX and 53BP1.

We thank the referee for this comment, which prompted us to perform new experiments to confirm our results showing that Tau knockdown increases endogenous DNA damage. Indeed, to provide additional evidence that Tau depletion favors DNA double-strand breaks (DSBs), we used an immunofluorescence-based assay to detect the frequency of 53BP1 foci, an established mediator of DNA DSB repair (Lukas *et al.*, 2011). As shown in Fig. 3a and 3d of the revised version of the manuscript, the frequency of cells with more than 20 53BP1 foci was significantly higher for cells with Tau depletion than for those without Tau depletion, confirming our finding that Tau depletion alters genetic stability by increasing endogenous DNA damage.

Our results showing that simultaneous Tau and CDA depletion aggravates genetic instability by increasing the frequency of γ -H2AX foci to levels greater than observed when each protein is depleted individually (Fig. 6a-b) led us to test whether the knockdown of CDA alone or together with Tau enhanced 53BP1 focus formation. As shown in Fig. 6c of the revised manuscript, the knockdown of CDA in control cells generated nuclei with a larger number of 53BP1 foci. Thus, CDA loss was sufficient to induce DNA damage and replication stress. The simultaneous knockdown of Tau and CDA caused a shift towards nuclei with the presence of larger numbers of 53BP1 foci in nuclei, indicating that the simultaneous depletion of both proteins further stimulates DSB repair.

These results have been added to the *Results* section, page 8 lines 181-187 and page 15 lines 351-361 as follows:

Page 8 lines 181-187

“We thus analyzed the impact of Tau knockdown on phosphorylated H2AX (γ -H2AX) levels, **53BP1 focus formation** and UFB frequency (Fig. 3a-e). Tau silencing increased γ -H2AX levels, as revealed by WB blots and the high percentage of cells with more than 20 γ -H2AX foci (Fig. 3b-c). Tau knockdown also **generated a significantly higher frequency of 53BP1 foci in the nuclei**, and resulted in a significant increase in UFB frequency, consistent with the existence of **endogenous DNA damage and replication stress** in Tau-deficient cells (Fig. 3d-e).”

Page 15 lines 351-361

“Finally, we explored how simultaneous depletion of the CDA and Tau proteins led to a synthetic lethal interaction. In previous studies, we showed that CDA deficiency led to spontaneous DNA damage, as revealed by the constitutive activation of γ -H2AX, and an increase in UFB frequency¹¹. We show here that Tau deficiency also results in increases in γ -H2AX levels, **the number of 53BP1 foci** and UFB formation. Moreover, an increase in R-UFB formation was observed following the depletion of either CDA or Tau protein. We therefore monitored these markers after simultaneous Tau and CDA downregulation. As expected, in control cells, the depletion of CDA alone led to increases in γ -H2AX levels, **the number of 53BP1 foci**, and the prevalence of both total UFBs and R-UFBs (Fig. 6a-e). Moreover, CDA depletion led to an increase in Tau mRNA and protein levels (Fig. 6a and Supplementary Fig. 6a). Finally, downregulating CDA expression in Tau-deficient cells led to **a marked shift towards the presence of larger numbers of γ -H2AX and 53BP1 foci in nuclei and an increase in the numbers of total and rDNA-associated UFBs (Fig. 6a-e)**. Thus, the simultaneous depletion of both proteins aggravated genome instability.”

The *Discussion* section page 16 lines 388-391 and page 17 lines 401-403 has been modified as follows:

Page 16 lines 388-390

“H2AX phosphorylation and 53BP1 recruitment events have emerged as highly specific and sensitive molecular markers for the monitoring of DNA damage initiation and repair⁵⁷. In Tau-deficient HeLa cells, high levels of γ -H2AX and large numbers of 53BP1 foci were observed, indicating an early cellular response to the induction of DSBs.”

Page 17 lines 401-403

“A deficiency of both Tau and CDA resulted in particularly high levels of γ -H2AX and large numbers of 53BP1 foci, and a higher prevalence of UFBs.”

Representative images of immunofluorescence labeling for γ -H2AX and 53BP1 foci have been added in Fig. 3a of the revised version of the manuscript.

5. Fig.3j: Run a RT- control to ensure that no genomic DNA contamination occurred.

We apologize for not clarifying this point earlier in the manuscript. To ensure that no genomic DNA contamination occurred during reverse transcription, we performed an on-column DNase digestion while purifying RNAs. We also carried out the reverse transcription step of RT-qPCR experiments in the absence of reverse transcriptase. We observed no 45S pre-rRNA amplification in the RT-negative control samples (Fig. 3a-b for the reviewer only), firmly demonstrating the absence of genomic DNA contamination in the RNA preparation.

Figure 3 | for the reviewer only: Amplification curves for 45S pre-rRNA gene in normal RT-qPCR conditions (denoted HeLa-Ctrl_(Tau) in the table) and RT-negative control conditions (denoted RT Neg). As shown on the amplification curves (Fig. 3a, red arrow), samples corresponding to the RT-negative samples displayed no amplification, even after 40 cycles of RT-qPCR (Fig. 3b).

The *Results* section page 10-11 lines 245-247 has been modified accordingly, as follows:

“We measured pre-rRNA levels in control cells and in cells with stable or transient Tau depletion. Control qPCR assays on RNA samples were carried out in parallel, in the absence of reverse transcriptase, to check that the amplification observed did not result from the presence of contaminating genomic DNA (data not shown).”

Reviewer #2 (Remarks to the Author):

Title: New functions for Tau protein in maintaining ribosomal DNA stability and cytidine deaminase-deficient cell survival

Authors: Bou Samra et al.

In this manuscript by Samra et al., the authors have tried to understand the mechanism(s) which allow the survival of cells which lack BLM and/or CDA. Using a range of cell biology based experiments; the authors indicate that a synthetic lethal interaction existed between CDA and Tau in cells which lack BLM and/or CDA. Presence of Tau was necessary for the ribosomal DNA (rDNA) stability. Hence absence of Tau led to the formation of a new class of ultrafine anaphase bridges, which contributed to genome instability along with CDA. Finally the authors postulate a role of Tau in ribosomal RNA synthesis.

The manuscript is novel, generally well written with defined controls. The genetic screen carried out by the authors is appreciated by this reviewer. However the cell biology experiments done to test the hypothesis require additional experiments and validation, before the manuscript can be considered for publication in Nature Communications.

We appreciate the reviewer's positive comments concerning our genetic screen for synthetic lethality and the novelty of our work. We provide a point-by-point response to the reviewer's individual comments below.

Major critiques:

1. The authors need to find a defined mechanism by which Tau affects ribosomal DNA stability. While the data at this stage strongly suggests that Tau affects indeed genetic/genomic stability, it does not provide compelling evidence in favor of the mechanism(s) necessary for the process. For example, how does Tau regulate rDNA transcription?

We thank the reviewer for this comment, which prompted us to perform new experiments to answer this question.

We show in the manuscript that stable and transient Tau depletion significantly decreased *45S* pre-rRNA synthesis (Fig. 3n of the revised version of the manuscript). CDA-deficient cells overexpressing Tau had higher levels of *45S* pre-rRNA (Fig. 3o of the revised version of the manuscript). These data suggest that Tau expression may regulate rDNA transcription. UBTF is a transcription factor required for Pol I-mediated rRNA synthesis. Given the colocalization of nucleolar Tau and UBTF and their similar occupancy profiles on rDNA repeats, we hypothesized that the lower levels of pre-rRNA synthesis in Tau-depleted cells might be linked to a decrease in UBTF recruitment to rDNA loci. We thus performed ChIP-qPCR assays with the Tau-1 and UBTF antibodies. Tau downregulation led to a significant decrease in Tau binding to rDNA promoter regions, as expected (Fig. 3p of the revised manuscript). Moreover, Tau downregulation also reduced UBTF binding to rDNA promoters without affecting total UBTF protein levels (Fig. 3q and supplementary Fig. 3k of the revised manuscript). These results indicate that Tau regulates rDNA transcription through UBTF recruitment to rDNA promoter regions.

These results have been added to *Results* section, pages 11 lines 251-259 as follows:

“As Tau expression was correlated with pre-rRNA levels, we investigated the mechanism by which the decrease in Tau levels decreased 45S pre-rRNA synthesis, by performing ChIP-qPCR assays to analyze the effect of Tau downregulation on UBTF recruitment to rDNA repeats. We found that Tau downregulation halved nucleolar Tau binding to rDNA repeats in Tau-depleted cells, as expected (Fig. 3p). Moreover, Tau downregulation also significantly reduced the recruitment of UBTF to the promoter regions of rDNA units (Fig. 3q), without affecting total UBTF protein levels (Supplementary Fig. 3k). Thus, Tau regulates rRNA synthesis through the recruitment of UBTF to rDNA promoters.”

The *Abstract* page 2 lines 29-31 and the *Discussion* section page 17-18 lines 419-423 and page 18 lines 426-428 have been modified accordingly, as follows:

Page 2 lines 29-31

“We also found that Tau is recruited, along with upstream binding factor (UBTF), to rDNA loci. Tau downregulation decreased UBTF recruitment, rRNA synthesis, ribonucleotide levels, and affected rDNA stability, leading to the formation of a new subclass of human ribosomal ultrafine anaphase bridges.”

Pages 17-18 lines 419-423

“As Tau and UBTF were colocalized at nucleolar sites and Tau downregulation impaired the recruitment of UBTF to rDNA promoter regions, it was tempting to speculate that UBTF loss was also synthetic lethal with CDA deficiency. UBTF knockdown seemed to favor DNA

damage and genomic instability⁶⁶, but our data demonstrated that UBTF downregulation was not synthetic lethal with CDA deficiency.”

Page 18 lines 426-428

“As nucleolar regions contain rRNA genes, we propose a putative role for Tau in rRNA synthesis. Indeed, we found that the downregulation of Tau expression decreased levels of 45S pre-RNA, by affecting the recruitment of UBTF to rDNA repeats, consistent with a role for Tau in the regulation of rRNA synthesis.”

2. The authors again do a very thorough job establishing the parameters required for the formation of R-UFB. However there are some inherent contradictions which need to be addressed. If the lack of CDA increases Tau (as the authors have repeatedly shown) – then what is the physiological basic of concomitant silencing of both Tau and CDA?

We apologize for not clarifying this point in our initial manuscript.

Synthetic lethality is based on the combination of deficiencies in the expression of two (or more) genes to induce cell death, whereas a lack of expression for only one of the genes concerned does not affect cell survival. Synthetic lethality approaches are gaining ground in the development of cancer-specific therapies. In our work, the concomitant silencing of Tau and CDA provides a physiological basis for the development of new anti-cancer therapies. Indeed, using a synthetic lethality approach, we showed that the concomitant depletion of CDA and Tau impaired the clonal growth of cells and was accompanied by an increase in genetic instability. We also showed that a complete absence or low levels of CDA was associated with higher levels of Tau mRNA and protein, and vice versa. As mentioned in the

discussion, several reports have suggested that Tau levels may have prognostic or predictive value in some cancers, due to their effects on the efficacy of microtubule-targeted therapies and patient care. In parallel, CDA is increasingly being seen as important, based on our finding that a deficiency of this protein significantly increases sensitivity to aminoflavone treatment (Mameri *et al.*, 2016). We therefore suggest that targeting Tau in CDA-deficient cells or targeting CDA in Tau-deficient cells may be useful approaches, guiding treatment choices and improving the efficacy of targeted therapies.

We have clarified this point by replacing the following sentences in the *Discussion* section pages 19-20 lines 470-472:

“Thus, taking expression status into account simultaneously for the Tau and CDA proteins should make it possible to guide treatment choices, improving the efficacy of targeted therapies”.

with:

“Thus, we suggest that targeting Tau in CDA-deficient cells or targeting CDA in Tau-deficient cells may be a useful option, guiding treatment choices and increasing the efficacy of targeted therapies”.

3. This reviewer is of the viewpoint that the *in vivo* link between the transcript levels of CDA and MAPT (Tau) in patient cell lines and tissues is very interesting. This should be the final figure of the paper. Additionally the authors should consider supplementing the *in vivo* data with mouse xenograft experiments using HeLa stable lines overexpressing and/or ablating BLM, Tau and CDA, individually and in combination.

We thank the reviewer for these interesting suggestions.

To address the first part of the comment, we have shifted the paragraph concerning the correlation between the levels of *CDA* and *MAPT* transcripts in cancer cell lines and tissues to the end of the *Results* section, in an additional section entitled “***CDA and MAPT transcript levels are negatively correlated in cancer cell lines and tissues***”. Moreover, to consolidate our findings, we analyzed additional *in silico* *CDA* and *MAPT* mRNA data from large-scale gene expression data sets (microarrays and mRNA sequencing), retrieved from the cBio cancer genomics portal (Cerami *et al.*, 2012 and Gao *et al.* Sci. Signal. 2013). We carried out Pearson correlation analyses to assess the strength of the association between *CDA* and *MAPT* transcript levels for all datasets analyzed, as routinely performed for analyses of this kind. We found a negative correlation between *CDA* and *MAPT* transcript levels in the Cancer Cell Line Encyclopedia (CCLE; Barretina *et al.* 2012) cohort of 967 cancer cell lines (Fig. 7b). Furthermore, an analysis of The Cancer Genome Atlas (TCGA) transcriptomic data sets for cancer tissues also showed a significant inverse correlation between levels of *CDA* and *MAPT* gene expression, particularly in papillary and clear cell kidney carcinomas (Fig. 7d-e), prostate adenocarcinoma (Fig. 7f) and in pheochromocytoma and paraganglioma (Fig. 7g).

These results have been added to the *Results* section, page 15-16 lines 363-381 as follows:

“*CDA and MAPT transcript levels are negatively correlated in cancer cell lines and tissues*

BS cells (BLM⁻/CDA⁻) and CDA-deficient HeLa cells overexpressed Tau, as described above. Tau levels have been reported to increase in certain subtypes of breast cancer, leading to resistance to microtubule-targeting drugs⁵¹. We have recently shown that CDA expression is downregulated in ~60% of cancer cells and tissues⁵². We therefore investigated the possible correlation between CDA loss and Tau overexpression in physiological conditions. We did this by *in silico* analyses of *MAPT* and *CDA* expression levels in several cohorts of cancer cell

lines and tissues (refer to the *Methods* section for data sets). We carried out Pearson's correlation analyses to assess the strength of association between *CDA* and *MAPT* transcript levels. An analysis of two cohorts of cancer cell line samples^{52,53} showed that *MAPT* levels were significantly higher in samples with low *CDA* levels, and vice versa (Fig. 7a-b). We then compared *CDA* and *MAPT* mRNA levels in gene expression data sets for different tissue samples⁵⁴⁻⁵⁶ (refer to the *Methods* section for link). The negative correlation between *CDA* and *MAPT* transcript levels was significant in cancers of several tissues, including breast cancer (Fig. 7c), papillary and clear cell kidney carcinomas (Fig. 7d-e), prostate adenocarcinoma (Fig. 7f), and pheochromocytoma and paraganglioma (Fig. 7g). Overall, our data indicate the existence of a causal link between the expression of the *CDA* and *MAPT* genes that could be exploited in the development of new anti-cancer strategies.

We have also shifted the corresponding paragraph in the *Discussion* section to the end of this section, just before the general conclusion.

With the aim of providing support for the *in vivo* link between levels of *CDA* and *MAPT* (*Tau*) transcripts in patient cell lines and tissues, we followed the reviewer's second suggestion and performed a series of xenograft experiments in mice with the HeLa-Ctrl_(Tau) and HeLa-shTau cell lines after obtaining regulatory and ethics approval required for such studies in mice (one month). Cells from these two cell lines (5×10^6 cells per xenograft) were subcutaneously (s.c) implanted into nude mice (12 mice per cell line). Tumor size was monitored every 3 to 4 days. Once the tumor reached a volume of 200 mm^3 , mice were randomized to two groups for intra-tumor treatment with a pool of control or *CDA* siRNAs (refer to *Methods* section for siRNA sequences) administered with the AteloGene kit (Koken, Japan), according to the manufacturer's instructions. Transfections were repeated twice, 72 h apart. At the end of the transfections, tumors were collected and analyzed.

We encountered several problems while setting up the experiments:

- HeLa-shTau cells grew slower than control cells. As shown in Fig. 1a-b for the reviewer only, only two Tau-depleted tumors (16.7%) reached the specified volume of 200 mm³ after 28 days of cell engraftment. By contrast, 75% of control tumors had already reached this volume by the same time point (Fig. 1a-b).
- Despite the use of the AteloGene kit (Koken, Japan) to transfect tumors with control and CDA siRNAs, we were unable to downregulate CDA expression.

These preliminary results demonstrated the need for further technical adjustments for the *in vivo* xenograft experiments. It will take us a few more months to perform these experiments. Furthermore, we feel that the data that these experiments would generate, although very interesting *per se*, would be unlikely to modify the message of this article, particularly as the analyses presented in Fig. 7 clearly demonstrate the *in vivo* link between the levels of CDA and Tau transcripts in tumor tissues from patients.

Figure 1 | HeLa-Ctrl_(Tau) and HeLa-shTau cells (5×10^6) were implanted s.c. in nude mice (12 mice per group). Tumor volumes were measured on day 13 and every 3-4 days thereafter and plotted against time. **(a)** Mean tumor volumes for 12 tumors shown as a single point for each time point. **(b)** Each tumor volume plotted individually for each time point.

4. Figure 1E, 2A, 2D – the decrease in the levels of Tau is consistently quantitated in Figure 2B, 2E to around 50%. However the blots do not seem to reflect this. Blots should be redone with lesser amount of lysates or lower exposures provided. Why are SD/SEM not shown in Figure 2B, 2E?

As suggested by the reviewer, we have addressed this point by providing a shorter exposure for each of the immunoblots with Tau-1 in Fig. 2c and Supplementary Fig. 2d and 2f of the revised version of the manuscript. These new exposures clearly show the lower levels of Tau protein in the following conditions:

- After BLM complementation (Fig. 2c and Supplementary Fig. 2d of the revised manuscript), inducing the expression of CDA
- After CDA complementation (Supplementary Fig. 2f of the revised manuscript)
- After Tau depletion with Tau-specific siRNAs (Supplementary Fig. 2d and 2f of the revised manuscript).

We did not show SD/SEM values in the initial version of the manuscript because we performed a relative quantification of Tau-1 immunoreactivity only for the corresponding immunoblots (Supplementary Fig. 2d and 2f). In the revised version of the manuscript, relative quantifications of Tau-1 are now shown as mean values for independent experiments in Supplementary Fig. 2e and 2g. Thus, using Tau-1 antibody, BLM complementation in BS cells reduced Tau levels by 36.5% (Fig. 2d) and 47.5% (Supplementary Fig. 2d). CDA

expression in BS cells reduced Tau levels by 39% (Supplementary Fig. 2g). Tau depletion decreased Tau levels by 45% and 51% in BS-Ctrl_(BLM) and BS-Ctrl_(CDA) cells (Supplementary Fig. 2e and 2g, black bars), respectively, and by 54% and 33% in BS-BLM and BS-CDA cells (Supplementary Fig. 2e and 2g, white bars), respectively. The error bars indicate the SEM.

The *Results* section page 7 lines 155-156 and lines 162-163 has been modified accordingly, as follows:

Page 7 lines 155-156

“However, a 45-48% decrease in Tau protein levels resulted in a 60% decrease in the colony-forming ability of BLM-null cells (Fig. 2g-h and Supplementary Fig. 2e), confirming the observations made in the primary shRNA screen.”

Page 7 lines 162-163

“The stable expression of CDA in BS cells reduced Tau protein levels by 35% and 39%, as detected by Tau-5 and Tau-1 antibodies, respectively (Fig. 2i-j and Supplementary Fig. 2f-g).”

5. Result (Line 160, 161) – to a level similar to that obtained by BLM complementation – can such a statement be made for different blots with possibly different exposures?

We agree with the referee that it is not possible to make such a statement for different blots with different exposure times. As indicated in our response to the previous point (point 4), we quantified Tau protein levels detected with tau 1 and we show quantifications of Tau levels as mean values for 3 independent experiments in Supplementary Fig. 2e and 2g. BLM and CDA complementation in BS cells reduced Tau protein levels by 47.5% and 39%, respectively.

As the decreases were not identical and to avoid confusion, we preferred to remove this sentence and to modify the *Results* section, page 7 lines 162-163 as follows:

“The stable expression of CDA in BS cells reduced Tau protein levels by 35% and 39%, as detected by Tau-5 and Tau-1 antibodies, respectively (Fig. 2i-j and Supplementary Fig. 2f-g).”

6. The authors should provide an explanation why CDA and UBTF were not detected in the screen (Figure 1A).

We apologize for not being clear enough in our initial explanations as to why CDA could not be detected in the screen. As mentioned in the *Results* section, the synthetic lethality screen was performed on BS-Ctrl_(BLM) cells, which lack both BLM and CDA, and on their BLM-complemented counterparts (BS-BLM), which express BLM and, thus, CDA. As BS-Ctrl_(BLM) cells already lack CDA, the genome-wide RNAi screen made it possible to identify genes enabling BLM- and/or CDA-deficient cells to survive despite constitutive DNA damage and replication stress. This is why CDA was not detected in the screen.

We have clarified this point, by adding the following to the *Results* section page 5 lines 97-98:

“This screen might, therefore, allow the identification of genes required for the survival of BLM- and/or CDA-deficient cells”.

We fully agree with the reviewer’s comment about UBTF. Indeed, as UBTF colocalized with Tau to nucleolar regions, and the amount of this protein recruited to rDNA promoters depended on the amounts of Tau recruited to the same regions, it is tempting to speculate that

UBTF loss might also be synthetic lethal with CDA deficiency, and that its presence should have been detected in the screen.

We addressed this point, by determining whether UBTF depletion in BS-Ctrl_(BLM) cells was synthetic lethal with BLM and/or CDA deficiencies. We transfected BS-Ctrl_(BLM) and BS-BLM cells with a pool of four UBTF-targeting siRNAs. Forty-eight hours after transfection, cells were monitored for colony formation (Supplementary Fig. 3l-m). UBTF downregulation did not affect the clonal growth of BS-Ctrl_(BLM), nor BS-BLM cell lines (Supplementary Fig. 3m), so UBTF depletion was not synthetic lethal with BLM and/or CDA deficiencies. This explains why we did not detect UBTF in the genetic screen.

The *Results* section page 11 lines 260-268 has been modified accordingly, as follows:

“Our results showing UBTF and Tau colocalization in nucleolar regions and at rDNA repeats led us to hypothesize that, like Tau deficiency, UBTF deficiency might present a synthetic lethal interaction with CDA deficiency, albeit one not detected in the RNAi screen. We therefore investigated whether UBTF downregulation impaired the survival of CDA-deficient cells. We transfected BS-Ctrl_(BLM) and BS-BLM cells with UBTF-targeting siRNAs (Supplementary Fig.3f). Then, 48 hours after transfection, both cell lines were monitored for colony formation. UBTF downregulation had no effect on clonal growth in either cell line (Supplementary Fig. 3g), demonstrating an absence of synthetic lethal interaction between UBTF and BLM and/or CDA deficiencies.”

The *Discussion* section page 17-18 lines 419-423 has been also modified as follows:

“As Tau and UBTF were colocalized at nucleolar sites and Tau downregulation impaired the recruitment of UBTF to rDNA promoter regions, it was tempting to speculate that UBTF loss

was also synthetic lethal with CDA deficiency. UBTF knockdown **seemed** to favor DNA damage and genomic instability⁶⁷, but **our data demonstrated that UBTF downregulation was not synthetic lethal with CDA deficiency.**”

7. The authors repeatedly show that decrease in the level of CDA causes an increase in the levels of Tau? What is mechanism for this to happen?

We thank the referee for raising this interesting point. Our results clearly show that CDA levels modulate Tau expression. A lack of CDA in BS cells (Fig. 2a-e of the revised version of the manuscript), and stable or transient CDA depletion in HeLa cells (Fig. 2n, Fig. 3j and Fig. 6a of the revised manuscript) increase Tau expression. By contrast, CDA complementation in BS cells decreased Tau levels (Fig. 2i-j and Supplementary Fig. 2f-g of the revised manuscript). We previously reported that CDA downregulation in cancer cells was mediated by DNA methylation (Mameri *et al.*, 2017). As shown in Fig. 2 for the reviewer only, the treatment of HeLa cells with the DNA methyltransferase activity inhibitor 5-Aza-2'-deoxycytidine (5-Aza-dC), resulting in DNA methylation, induced an increase in CDA levels and a decrease in Tau levels. Thus, whatever the origin of CDA overexpression — exogenous expression (BS-CDA cells, Fig. 2i) or endogenous CDA induction through the treatment of cells with 5-Aza-dC (Fig. 2 below) — it leads to the downregulation of Tau expression. Similarly, whatever the origin of CDA downregulation — physiological (BS cells, Fig. 2a-g), or mediated by RNA interference (HeLa cells, Fig. 2n, Fig. 3j and Fig. 6a of the revised version of the manuscript) — it led to Tau upregulation. We are currently investigating the mechanism underlying the regulation of Tau expression by CDA. However, it will take several months to answer this question, which we feel is, in any case, beyond the specific scope of this manuscript.

Figure 2 | for the reviewer only: Tau-1, DNMT1, CDA and GAPDH protein levels assessed by immunoblotting. HeLa cells were left untreated or treated with 1 μ M 5-Aza-dC for 96 hours. DNMT1 was used as a control, because DNMT1 levels decrease in response to 5-Aza-dC treatment. GAPDH was used as a loading control.

8. Figure 3D-3G – the drug sensitivity assays should be done with two other isogenic lines HeLa-shBLM and HeLa-shCDA and data for all four lines be presented together.

As suggested by the referee, we monitored drug sensitivity in the isogenic cell lines HeLa-Ctrl_(CDA) and HeLa-shCDA. For the sake of clarity and to be consistent with the logic of the experiments, these results have been added to the revised manuscript as Supplementary Fig. 3c-g and Fig. 3k.

The *Results* section, page 9 lines 197-207, has been modified accordingly, as follows:

“As described above, CDA-deficient BS cells overexpressed Tau. Consistent with this finding, HeLa cells with physiological levels of BLM but a stable depletion of CDA also overexpressed Tau (Fig. 3j). We subjected these cells to drug sensitivity assays. CDA-deficient cells and control cells had similar sensitivities to ICRF-159, α -amanitin and APH (Supplementary Fig. 3c-e). After treatment with CPT or HU, CDA-deficient cells became slightly less or more sensitive than control cells, respectively (~1.5-fold; Supplementary Fig. 3f-g). By contrast, CDA depletion led to a 3.5-fold increase in resistance to CX-5461 (Fig. 3k), probably due to Tau overexpression. The greater susceptibility of Tau-deficient cells to CX-5461 treatment was thus associated with a high resistance of Tau-overexpressing CDA-

deficient cells to the same treatment. Collectively, these findings provide further support for a role for Tau in rRNA transcription.”

We also assessed the sensitivity of the isogenic cell lines HeLa-Ctrl_(BLM) and HeLa-shBLM to different cytotoxic agents, as requested by the reviewer. However, as we demonstrated an absence of involvement of BLM deficiency in the synthetic lethal interaction between CDA and Tau deficiencies (Fig. 2j and 2n of the revised version of the manuscript), we decided, for the sake of clarity, not to include these data in the manuscript. We therefore communicate the results below for the reviewer only (Fig. 3a-e, for the reviewer only). BLM-deficient cells and control cells were similarly sensitive to APH, CPT, HU, CX-5461 and ICRF-159 (Fig. 3a-e), and BLM-deficient cells were slightly more resistant to α -amanitin than control cells (Fig. 3f).

Figure 3 | For reviewer only: HeLa-Ctrl_(BLM) and HeLa-shBLM cells were exposed, for 72 h, to aphidicolin **(a)**, camptothecin **(b)**, hydroxyurea **(c)**, CX-5461 **(d)**, ICRF-159 **(e)** and α -amanitin **(f)**. Each data point is the mean of at least three independent experiments performed in triplicate. Error bars represent the SEM.

9. Figure 3I – to possibly determine whether Tau enhances the binding of UBTF to the rDNA (see comment 1 above), the authors should try to do a reciprocal re-CHIP using UBTF and CDA antibodies.

We have already clarified this point in our response to comment 1. Using ChIP-qPCR assays, we showed that the amount of UBTF recruited to the rDNA promoter regions depended on the level of Tau expression. Indeed, Tau downregulation significantly reduced UBTF recruitment to rDNA loci (Fig. 3p-q of the revised version of the manuscript).

10. The authors should try to co-localize Tau with UBTF at the termini of R-UFB. This alone (and not co-staining with either CREST and/or PICH) should be a criteria to define R-UFBs.

We apologize for any confusion and for not clarifying this point in the initial version of the manuscript. Ultrafine anaphase bridges (UFBs) are “thread-like” structures that form a bridge linking the daughter DNA masses separating during anaphase. UFBs have been shown to contain DNA, but they cannot be stained with conventional DNA dyes or antibodies against histones (Baumann *et al.*, 2007; Chan *et al.*, 2007). They can be visualized by detection of the helicase-like protein, PICH (Plk1-interaction checkpoint “helicase”) (Baumann *et al.*, 2007). Other proteins, such as the RecQ helicase BLM, can also be recruited to bridges, but only to pre-existing PICH-coated UFBs (Chan *et al.*, 2007). Thus, the total population of UFBs can be visualized only by PICH staining. PICH-coated UFBs frequently link centromeric loci (Baumann *et al.*, 2007). In the manuscript, we used CREST and PICH antibodies to track UFBs derived from centromeric DNA. Similarly, we tracked rDNA-associated UFBs (R-

UFBs) by labeling for UBTF and PICH, or Tau-1 and PICH; Tau-1 and UBTF labeled rDNA repeats. Thus, double-staining with Tau-1 and UBTF (without also staining for PICH), as suggested by the reviewer, would not detect the presence of R-UFBs. This is well illustrated in Fig. 31 of the revised version of the manuscript showing a labeling of anaphase cells with Tau-1 and UBTF antibodies. Tau-1 and UBTF clearly stained rDNA repeats. However, it was not possible to detect R-UFBs in the absence of costaining with the PICH antibody.

Thus, for sake of clarity and to avoid confusion, we have modified the *Results* section page 12 lines 286-292 as follows:

“Different types of UFBs have been described in human cells, extending from centromeres (C-UFBs), common fragile sites (CFS-UFBs), or telomeres (T-UFBs)³⁸⁻⁴⁰. UFBs cannot be stained with conventional DNA dyes or antibodies against histones^{40,41}. However, they can be visualized by detection of the helicase-like protein PICH (Plk1-interaction checkpoint “helicase”)⁴¹. PICH is a member of the SNF2 family of proteins; it has dsDNA translocase activity, and decorates UFBs along their entire length. Other proteins, such as the RecQ helicase BLM, can also be recruited to bridges, but only to pre-existing PICH-coated UFBs⁴⁰. Thus, the total UFB population can be detected only by staining for PICH.”

11. A schematic model which indicates the inter-relationship and genetic interactions involving BLM, CDA and Tau needs to be shown.

A schematic model indicating the inter-relationship and genetic interactions involving CDA and Tau has been added as Fig. 8 to the revised version of the manuscript. As previously

mentioned, we prefer not to link BLM to this model since we clearly confirmed that all our findings were directly linked to CDA and/or Tau deficiencies.

Figure 8 is now cited at the end of the discussion section, page 20, line 478.

Other critiques:

1. Abstract, line 28 – It should be Tau is overexpressed in CDA/BLM deficient cells.

We demonstrated that stable or transient depletion of CDA induced Tau overexpression, independently of BLM expression status (Fig. 3j and Fig. 6a). Thus, to avoid confusion, we prefer not to link BLM to Tau overexpression.

2. Since the role of UBTF is quite important in the context of the paper, it should be mentioned in the abstract.

The abstract page 2 lines 29-32 has been modified as follows:

“We also found that Tau is recruited, along with upstream binding factor (UBTF), to rDNA loci. Tau downregulation decreased UBTF recruitment, rRNA synthesis, ribonucleotide levels, and affected rDNA stability, leading to the formation of a new subclass of human ribosomal ultrafine anaphase bridges.”

3. Introduction (line 69-70); The BS cells are susceptible to increase in cell lethality due to depletion....

The *Introduction* section page 3 lines 68-69 has been modified as follows:

“The BS cells were likely to display higher levels of cell lethality due to the depletion of the microtubule-associated protein Tau.”

4. Introduction (line 76) – it is not co-localization, it is co-recruitment.

The *Introduction* section page 4 lines 75-78 has been modified as follows:

“We observed the corecruitment of Tau and UBTF to the nucleolar organizing regions (NORs), and found that Tau silencing reduced the recruitment of UBTF to rDNA repeats, thereby impairing ribosomal DNA (rDNA) transcription.”

5. Introduction (line 78, 79) – Tau bound (not interacted) with active rDNA repeats. However at present there is no evidence to suggest that this recruitment/binding favored rDNA transcription.

With our new results presented in the revised version of the manuscript (Fig. 3p-q), we can firmly conclude that Tau recruitment to rDNA regions regulates UBTF recruitment to these regions and, thus, rDNA transcription.

Accordingly, the *Introduction* section page 4 lines 75-78 has been modified as follows:

“We observed the corecruitment of Tau and UBTF to the nucleolar organizing regions (NORs), and found that Tau silencing reduced the recruitment of UBTF to rDNA repeats, thereby impairing ribosomal DNA (rDNA) transcription.”

6. Figure 3C – which UFBs are the authors counting here?

Fig. 3c from the original manuscript is now Fig. 3e in the revised manuscript. We used PICH staining to track UFBs, so the entire population of detectable UFBs is represented in Fig. 3e.

7. Figure 3H- The nucleolus is not very well defined in the DAPI image. So either a marker for nucleolus be co-stained or alternate photos given.

This figure is now Fig. 3I in the revised manuscript. An alternative image is now provided.

8. Figure 3L, M – the LC-MS/MS spectra should be put in as supplemental data.

The LC-MS/MS spectra are provided in Supplementary Fig. 3n-o and Supplementary Fig. 3r-s of the revised version of the manuscript.

9. Line 292 – the Figure mentioned as 3D, is actually 5D.

We apologize for this error, which has been corrected.

10. Line 318, 321 – Figure 6A-6C is actually 6A-6D.

We apologize for this error, which has been corrected.

Reviewers' Comments:

Reviewer #1:

Remarks to the Author:

The different points raised in the first review have been satisfactorily addressed. This very interesting and well-done work will greatly emphasize the importance of Tau protein.

Two minor points that still need to be addressed:

- Justify in the results (p6) the rationale to use Tau1 antibody in this study.
- Sup Fig.3h,i, note P-Thr231 and P-S396 instead of Thr231 and S396.

Reviewer #2:

Remarks to the Author:

Title: New functions for Tau protein in maintaining ribosomal DNA stability and cytidine deaminase-deficient cell survival

Authors: Samra et al.

The manuscript by Samra et al., the authors have tried to answer the queries raised by this reviewer. Almost all the queries have been successfully done or clarifications provided except the suggested experiment with mouse xenograft. Hence I believe the manuscript is much improved and is now almost conforms to the standards of Nature Communications.

I have a few minor suggestions which maybe attended to by the authors:

1. Figure 1A: according to this reviewer the box (which contains the three points indicating the assays which have been done) should come below the four plates with viability information but above the box indicating the number of candidates.
2. Figure 4, 5 and 6 should be merged into a single figure with some amount of data transferred to supplementary figures.
3. On reflection, more in depth statistical analysis of the patient data could have been done. For example a few points that could have been considered are (a) the sub-stratification of the patients based on stages of cancer and sub-stratification of the type of cancer (b) determining the area under the ROC curve (c) Kaplan Meyer analysis (d) COX regression analysis. These four analyses, especially for patients in TCGA and METABRIC (where the number of patients are higher) would be informative. Multiple datasets from TCGA and METABRIC should be analyzed. The patient IDs along with the raw numbers for the transcripts should be included as supplementary data. Importantly the authors should try to analyze whether in blood of the same patients there is a reciprocal correlation between the transcript levels of CDA and MAPT (such information is indeed available for patients in TCGA database)? Only then the effort may have a prognostic and diagnostic value.
4. Page 7, lines 162, 163: an explanation of the result or correlation with data is required.
5. Page 9: lines 204-206: the sentence can be rewritten.
6. Expand IGS (Page 10, line 242).
7. Page 10, lines 222, 223: delete "as previously reported" – there is no such information with respect to BS cells.

Reviewer #1 (Remarks to the Author):

The different points raised in the first review have been satisfactorily addressed. This very interesting and well-done work will greatly emphasize the importance of Tau protein.

Two minor points that still need to be addressed:

- Justify in the results (p6) the rationale to use Tau1 antibody in this study.

We thank the reviewer for this suggestion. We have modified the *Results* section page 6, lines 142-150 as follows (highlighted in yellow):

“The overexpression of Tau in BS cells was confirmed by reverse-transcription-quantitative PCR (RT-qPCR) and western blotting (WB) (Fig. 2a-c). We used Tau-5 monoclonal antibody, which detects all Tau proteins, regardless of their phosphorylation state (Fig. 2b). As we planned to investigate the nucleolar distribution of Tau in subsequent experiments and the Tau-5 antibody is not suitable for Tau nucleolar immunostaining, we also tested the only Tau antibody for which nucleolar staining has been reported²⁷, the Tau-1 antibody (Fig. 2c). This antibody recognizes the unphosphorylated Tau epitope Ser195-202. BLM expression reduced Tau mRNA levels by 59%, (Fig. 2a) and the amounts of protein detected by Tau-5 and Tau-1 by 28% and 36.5%, respectively (Fig. 2d-e).

- Sup Fig.3h,i, note P-Thr231 and P-S396 instead of Thr231 and S396.

We apologize for this error, which has been corrected.

Reviewer #2 (Remarks to the Author):

Title: New functions for Tau protein in maintaining ribosomal DNA stability and cytidine deaminase-deficient cell survival

Authors: Bou Samra et al.

The manuscript by Samra et al., the authors have tried to answer the queries raised by this reviewer. Almost all the queries have been successfully done or clarifications provided except the suggested experiment with mouse xenograft. Hence I believe the manuscript is much improved and is now almost conforms to the standards of Nature Communications.

I have a few minor suggestions which maybe attended to by the authors:

1. Figure 1A: according to this reviewer the box (which contains the three points indicating the assays which have been done) should come below the four plates with viability information but above the box indicating the number of candidates.

We thank the reviewer for this comment and we apologize for any lack of clarity in the existing version of Fig. 1a. The viability information is directly linked to the abundance of each gene in the various cell populations, as determined by barcoded sequencing. Hits were selected on the basis of their relative abundance with respect to selection criterion (the four plates with viability information). The box describing the three steps of the assay performed is therefore in the correct place.

For the sake of clarity, we have replaced the four plates displaying viability information with four boxes containing the same information (Fig. 1a of the revised version of the manuscript).

2. Figure 4, 5 and 6 should be merged into a single figure with some amount of data transferred to supplementary figures.

As suggested by the reviewer, we have merged Figs. 4, 5 and 6 into a single figure: Fig. 4 of the revised manuscript. We have also merged Supplementary Figs. 4, 5 and 6 to produce Supplementary Fig. 4 of the revised manuscript. In addition, we have modified the title of this new section, as follows:

Tau loss alters the genetic integrity of rDNA and CDA-deficient cells

3. On reflection, more in depth statistical analysis of the patient data could have been done. For example a few points that could have been considered are (a) the sub-stratification of the patients based on stages of cancer and sub-stratification of the type of cancer (b) determining the area ROC curve (c) Kaplan Meyer analysis (c) COX regression analysis. These four analyses, especially for patients in TCGA and METABRIC (where the number of patients are higher) would be informative. Multiple datasets from TCGA and METABRIC should be analyzed. The patient IDs along with the raw numbers for the transcripts should be included as supplementary data. Importantly the authors should try to analyze whether in blood of the same patients there is a reciprocal correlation between the transcript levels of CDA and MAPT (such information is indeed available for patients in TCGA database)? Only then the effort may have a prognostic and diagnostic value.

We thank the reviewer for these interesting suggestions. As suggested, we have included the patient IDs and raw numbers of transcripts in the revised version of the manuscript, in supplementary Table 6.

[Redacted text block]

[Redacted text block]

[Redacted text block]

[REDACTED]

[REDACTED]

[REDACTED]

[REDACTED]

[REDACTED]

4. Page 7, lines 162, 163: an explanation of the result or correlation with data is required.

We thank the reviewer for this comment. An explanation of the result has been added to the *Results* section, page 7 lines 167-168 as follows (highlighted in yellow):

“The stable expression of CDA in BS cells reduced Tau protein levels by 35% and 39%, as shown with the Tau-5 and Tau-1 antibodies, respectively (Fig. 2i-j and Supplementary Fig. 2f-g), suggesting that high levels of Tau expression in BS cells probably results from CDA deficiency rather than from BLM deficiency *per se*.”

5. Page 9: lines 204-206: the sentence can be rewritten.

As suggested by the reviewer, we have rephrased the sentence page 9, lines 209-211 as follows:

“This greater resistance of CDA-deficient cells to CX-5461 treatment was inversely correlated with the greater susceptibility of Tau-depleted cells to the same treatment.”

6. Expand IGS (Page 10, line 242).

IGS was mentioned earlier in the text, on page 10, line 242. At this position, IGS was defined as ‘intergenic spacer’.

7. Page 10, lines 222, 223: delete “as previously reported” – there is no such information with respect to BS cells.

We apologize for this error, which has been corrected.

Reviewers' Comments:

Reviewer #2:

Remarks to the Author:

None